# Large scale discovery of coronavirus-host factor protein interaction motifs reveals SARS-CoV-2 specific mechanisms and vulnerabilities

Thomas Kruse[1,9], Caroline Benz [2,9], Dimitriya H. Garvanska[1,9], Richard Lindqvist[3,4,9], Filip Mihalic[5], Fabian Coscia[1,8], Raviteja Inturi [5], Ahmed Sayadi[2], Leandro Simonetti [2], Emma Nilsson[3,4], Muhammad Ali [2], Johanna Kliche[2], Ainhoa Moliner Morro[6], Andreas Mund [1], Eva Andersson[5], Gerald McInerney [6], Matthias Mann[1], Per Jemth [5], Norman E. Davey[7], Anna K. Överby [3,4✉], Jakob Nilsson [1✉] & Ylva Ivarsson [2✉]

Viral proteins make extensive use of short peptide interaction motifs to hijack cellular host factors. However, most current large-scale methods do not identify this important class of protein-protein interactions. Uncovering peptide mediated interactions provides both a molecular understanding of viral interactions with their host and the foundation for developing novel antiviral reagents. Here we describe a viral peptide discovery approach covering 23 coronavirus strains that provides high resolution information on direct virus-host interactions. We identify 269 peptide-based interactions for 18 coronaviruses including a specific inter-action between the human G3BP1/2 proteins and an ΦxFG peptide motif in the SARS-CoV-2 nucleocapsid (N) protein. This interaction supports viral replication and through its ΦxFG motif N rewires the G3BP1/2 interactome to disrupt stress granules. A peptide-based inhibitor disrupting the G3BP1/2-N interaction dampened SARS-CoV-2 infection showing that our results can be directly translated into novel specific antiviral reagents.

[1] The Novo Nordisk Foundation Center for Protein Research, University of Copenhagen, Faculty of Health and Medical Sciences, Blegdamsvej 3B, 2200 Copenhagen, Denmark. [2] Department of Chemistry - BMC, Uppsala University, Box 576, Husargatan 3, 751 23 Uppsala, Sweden. [3] Department of Clinical Microbiology, Umeå University, 90185 Umeå, Sweden. [4] Laboratory for Molecular Infection Medicine Sweden (MIMS), Umeå University, 90186 Umeå, Sweden. [5] Department of Medical Biochemistry and Microbiology, Uppsala University, BMC, Box 582, Husargatan 3, 751 23 Uppsala, Sweden. [6] Department of Microbiology, Tumor and Cell Biology, Karolinska Institutet, Stockholm, Sweden. [7] Division of Cancer Biology, The Institute of Cancer Research, 237 Fulham Road, London SW3 6JB, UK. [8] Present address: Spatial Proteomics Group, Max Delbrück Center for Molecular Medicine in the Helmholtz Association, 13125 Berlin, Germany. [9] These authors contributed equally: Thomas Kruse, Caroline Benz, Dimitriya H. Garvanska, Richard Lindqvist. ✉email: anna.overby@umu.se; jakob.nilsson@cpr.ku.dk; ylva.ivarsson@kemi.uu.se

RNA viruses such as the Ebola, dengue, and coronaviruses cause a variety of diseases and constitute a continuous threat to public health. The coronaviruses are the largest single-stranded RNA viruses known and their genomic RNA encodes around 30 viral proteins[1]. During infection, each viral protein performs unique functions and interacts with a range of cellular protein host factors to allow viral proliferation and immune escape[2–5]. Precise disruption of viral-host factor interactions is an attractive strategy for developing novel antiviral reagents. The advantage of targeting these interactions is that resistance is less likely to develop and furthermore as the same host factor can be used by multiple viruses such reagents may provide broader spectrum activity. Numerous large-scale mass spectrometry (MS) based interaction screens[2,3,5], as well as CRISPR based screens[6–10] have been conducted to uncover host factor interactions and dependencies for SARS-CoV-2 allowing repurposing of drugs against human targets[11,12]. Although these methods have been transformative in our understanding of SARS-CoV-2 biology the molecular detail provided by these methods is not always sufficient to readily transform the results into novel antiviral reagents. Experimental approaches that would complement the existing powerful methods and provide a more detailed view of viral interactions with host factors could accelerate the development of new antivirals.

An attractive class of protein interactions that can be inhibited for therapeutic purposes are viral short linear interaction motifs (SLiMs) that bind to defined pockets on globular domains of the host factor[13,14]. SLiMs are short peptide motifs in unstructured regions of proteins and typically contain 2–4 amino acid binding determinants within a 10 amino acid stretch[15,16]. Viruses extensively use SLiMs to hijack cellular host factors and SLiMs can readily evolve through mutations in unstructured regions allowing viruses to interact with novel host factors[17–19]. Despite the importance of SLiMs for understanding viral biology, they are not uncovered by most current large-scale methods[15,16]. Proteomic peptide-phage display (ProP-PD) provides the opportunity to identify novel SLiM-based interactions and binding sites at high resolution[20]. As shown in a small scale pilot study on C-terminal peptides of viral proteomes, it can be used to faithfully capture SLiM-based host-pathogen interactions[21]. Here we describe a novel phage-based viral peptide library to map SLiMs from 23 coronaviruses mediating host factor interactions (Fig. 1a). This approach allows the simultaneous pan-viral identification of SLiM-based interactions with high resolution of the binding sites. We document the power of this approach by identifying novel SARS-CoV-2 specific SLiM mediated host factor interactions and directly translate our screening results into novel mechanistic insights and pinpoint a potential target for antiviral intervention.

## Results

**A pipeline for viral SLiM discovery**. We exploited recent developments of the ProP-PD technology[20,22] and established a pipeline to identify RNA virus SLiMs binding to host factors. Briefly, the pipeline consists of purifying protein domains and screening these against a novel phage display library displaying the unstructured parts of viral proteins. Following a number of selection cycles enriched phages are sequenced to identify the viral SLiMs binding a specific bait (Fig. 1a). We designed a unique phage display library (RiboVD library) that tile the unstructured regions of 1074 viral proteins from 229 RNA viruses, including SARS-CoV-2, SARS-CoV, MERS-CoV and 20 additional coronavirus strains. This library represents 19,549 unique 16 amino acid long peptides that are multivalently displayed on the major coat protein of the filamentous M13 phage. We scanned a

published host factor interactome for the SARS-CoV-2 viral proteins and recombinantly produced 57 domains from 53 cellular proteins reported to interact with SARS-CoV-2[2]. As transient SLiM-based protein interactions might be lost during purifications of viral proteins for subsequent mass spectrometry analysis, we screened an additional set of 82 peptide-binding domains. These domains were chosen because they were efficiently expressed and purified in *E. coli*[23] and at least 27 of them have previously been reported to act as viral-host factors and to be hijacked by SLiMs from viral proteins. In total, 139 recombinantly expressed and purified human bait proteins (Supplementary Data 1) were used in selections against the RiboVD library. Enriched phage pools were analyzed by next-generation sequencing (NGS) to identify viral peptides that bound to the bait. In light of the ongoing COVID-19 epidemic, we chose to focus on the interactions mediated by coronavirus proteins in the following, which represent 13% of the total number of interactions identified in the screen. Interaction data for the remaining RNA viruses will be published elsewhere.

We uncovered 269 putative SLiM-based interactions with 104 interactions identified using the 57 MS identified SARS-CoV-2 host factor baits and 165 interactions identified using the 82 peptide-binding domains. The interactions covers 44 domains from 42 human proteins and 64 viral proteins from 18 coronavirus strains (Supplementary data 2). Of these, 117 (43%) interaction pairs involved human coronavirus proteins. We validated 27 interactions using fluorescence polarization (FP) affinity measurements (Fig. 1b, Supplementary Fig. 1 and Supplementary Data 3). We visualized the information generated for human and bat coronavirus proteins in an extensive network (Supplementary Fig. 2). We also generated a map of the viral proteins mediating SARS-CoV-2, SARS-CoV and MERS-CoV interactions with human host factors (Fig. 1c). The map reveals common as well as unique interactions with host factors for these three coronaviruses. For instance, NSP14 of all three strains has a YxxL motif that binds to the clathrin coat adaptor protein AP2M1 with high affinity (Fig. 1b, Supplementary Fig. 1 and Supplementary Data 3), which may be linked to the trafficking of the viral protein or blocking of endocytosis of host proteins[24,25]. The N-terminal region of the E protein from all three strains binds to the FERM domains of Ezrin and Radixin via a recently established [FY]x[FILV] SLiM[26]. Interestingly, our data show that the FERM domains also bind to NSP3 of SARS-CoV and SARS-CoV-2, thus, they can be targeted by distinct viral proteins. The SARS-CoV NSP3 FERM binding site overlaps with a [FWY]xx[ILV] binding site for the ATG8 domains of the autophagy-related MAP1LC3A-C proteins. As an example of strain-specific interactions, we found that an N-terminal peptide from the Nucleocapsid (N) proteins from SARS-CoV-2 and SARS-CoV bound to the NTF2 domain of the homologous G3BP1 and G3BP2 proteins (G3BPs) with high affinity (Fig. 1b, c, Supplementary Fig. 1 and Supplementary Data 3). This N peptide contains an ΦxFG SLiM (where Φ is a hydrophobic residue) that resembles motifs in USP10 and UBAP2L and in the alphavirus nsP3 protein known to bind a hydrophobic pocket in the NTF2 domain of G3BP[27–30]. The ΦxFG SLiM is also present in the N proteins from bat beta coronaviruses and consistently the corresponding bat HKU5 peptide was identified in our screen (Supplementary Fig. 2 and Supplementary Data 2).

To pinpoint therapeutically relevant host protein-viral SLiM interactions we screened three of the identified peptide motifs for antiviral activity. To this end, we generated lentiviral vectors expressing GFP fused to four copies of one viral SLiM reasoning that this would inhibit binding of the corresponding full-length SARS-CoV-2 protein to the specific host factors through competition. As a control we used GFP fused to SLiMs containing

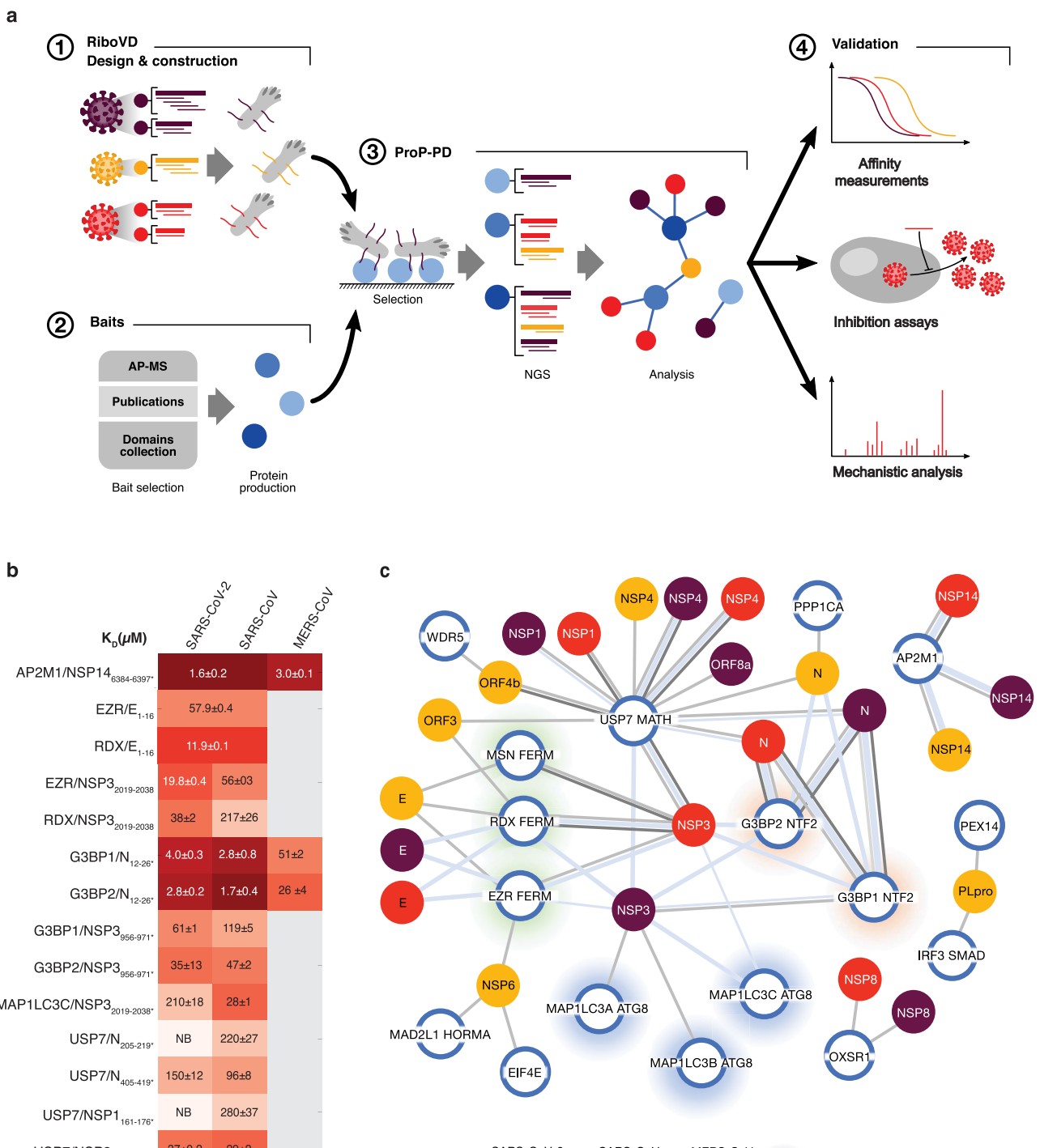

**Fig. 1 A pipeline for viral SLiM discovery. a** An overview of the platform for identifying viral SLiMs binding to cellular host factors. **b** $K_D$ values for the interactions between indicated viral peptides and host proteins. **c** Network of SLiM mediated interactions between the indicated viral proteins from SARS-CoV-2 (red), SARS-CoV (purple), and MERS-CoV (yellow) and cellular host factors (blue circles). Light grey connecting line indicates interactions validated by affinity measurements, the weight of the line represents the affinity of the interaction (thick, 1–10 μM; medium, 11–100 μM; thin, 101–500 μM). Dark grey lines indicate protein-protein interactions with additional evidence found in the other studies (Supplementary Data 2).

mutations in the binding motif. The host proteins targeted by viral peptides were G3BPs (SARS-CoV-2 N), Ezrin and Radixin (SARS-CoV-2 E and NSP3), and the MAP1LC3s (NSP3). VeroE6 cells were first transduced with the lentiviruses and 3 days later

infected with SARS-CoV-2 and viral titer determined after 16 h. This revealed that the G3BP-binding peptide from the N protein decreased viral titer 3.4-fold (Fig. 2a). To obtain a more potent inhibition of the SARS-CoV-2 N-G3BP interaction, we used a 25

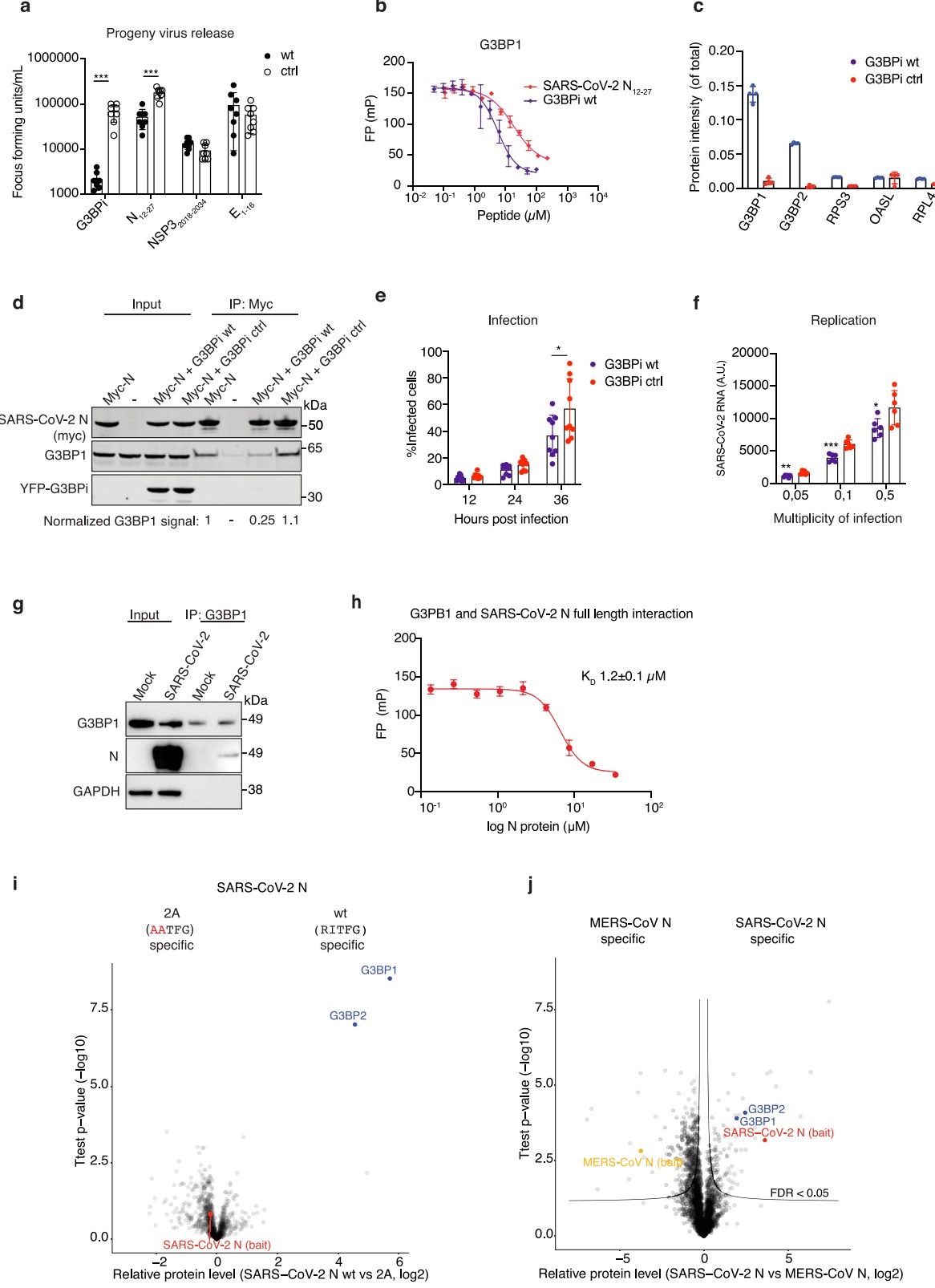

amino acid residue peptide from Semliki Forest virus (SFV) nsP3 containing two continuous FGDF like SLiMs that has previously been shown to bind G3PBs with high affinity[31]. Remarkably, this peptide binds approximately 10-fold stronger than the SARS-CoV-2 N peptide to both G3BP1 and G3BP2 ($K_D = 4 \mu M$ vs $K_D = 0.3 \mu M$, Fig. 2b and Supplementary Fig. 3a). We constructed "G3BP inhibitors" (G3BPi) by fusing sequences encoding one or

three copies of wild type (wt) or mutated (ctrl) SFV nsP3 SLiMs to GFP. As expected, mass spectrometry analysis confirmed that the major cellular targets of the G3BPi are the G3BPs (Fig. 2c, Supplementary Fig. 3b and Supplementary Data 4). Furthermore, expression of the G3BPi wt but not G3BPi ctrl prevented the binding of SARS-CoV-2 N to G3BP1 in cells (Fig. 2d). Consistent with these binding and competition data, lentiviral mediated

**Fig. 2 The interaction between N and G3BP1/2 is important for SARS-CoV-2 infection. a** Screen for SARS-CoV-2 antiviral activity of viral peptides including the G3BP inhibitor G3BPi ($p = 0.000013$) and $N_{12-27}$ ($p = 0.000005$). The amount of SARS-CoV-2 virus released was determined 16 h postinfection by focus forming assay ($n = 8$ independent experiments). **b** Affinity measurements of recombinant G3BP1 NTF2 binding to G3BPi and the SARS-CoV-2 N peptide ($n =$ two biological duplicates each containing three technical replicates). Shown is a representative plot from one of the experiments. **c** Quantitative mass spectrometry comparison of G3BPi wt and ctrl purified from HeLa cells ($n = 4$ technical replicates). **d** Purification of myc-tagged SARS-CoV-2 N expressed in HeLa cells and its interaction with G3BP1 analyzed by western blot. G3BPi wt or ctrl were co-expressed with myc-tagged N where indicated. Shown is a representative blot from three independent experiments. **e** Effect of G3BPi on % infected VeroE6 cells during 36 h of infection ($n = 9$ independent experiments) *$P = 0.0414$. **f** Amount of SARS-CoV-2 RNA measured 16 h postinfection with qPCR at different MOI in VeroE6 cells expressing G3BPi or control inhibitor ($n = 6$ independent experiments) *$P = 0.0291$, **$P = 0.0029$, ***$P = 0.0002$. **g** Endogenous G3BP1 was purified from mock or SARS-CoV-2 infected cells and probed for N. Shown is a representative blot from three independent experiments. **h** In vitro interaction of recombinant full-length SARS-CoV-2 N and the NTF2 domain of G3BP1 as measured by fluorescence polarization spectroscopy ($n =$ two biological duplicates each containing three technical replicates). Shown is a representative plot. **i** Quantitative mass spectrometry analysis of YFP-tagged SARS-CoV-2 N wt or 2 A purified from HeLa cells ($n = 4$ technical replicates). **j** Quantitative mass spectrometry analysis of YFP-tagged N SARS-CoV-2 and N MERS purified from HeLa cells ($n = 4$ technical replicates). Asterisks indicate statistical significance calculated by two-sided unpaired $T$ test (panel **a**, **e**, **f**). Mean ± SD indicated throughout in the graphs. Source data are provided as a Source Data file.

expression of the G3BPi in VeroE6 cells potently inhibited SARS-CoV-2 proliferation after 16 h of infection (Fig. 2a). An effect of the G3BPi was also evident in assays monitoring viral infection rates or replication (Fig. 2e, f). In a cell-based transfection assay monitoring assembly and release of virus-like particles mutating the G3BP-binding motif in N had no effect (Supplementary Fig. 3d). Thus, the approach presented here is useful for identifying important virus-host factor interactions that inhibit viral proliferation when disrupted.

**The N-G3BP1/2 interaction supports SARS-CoV-2 infection**. The above results prompted us to further investigate the N-G3BP interaction and its function during infection. The coronavirus N protein is important for viral replication, as well as packaging of the viral RNA[32–34]. The G3BPs are multi-functional RNA-binding proteins best known for their essential roles in innate immune signaling and the assembly and dynamics of cytosolic stress granules[35–38]. Stress granules are large protein-RNA assemblies formed in response to various stresses and viral infections[39–41]. The G3BPs have turned out to be major targets for viral interference and several viral proteins have been shown to recruit G3BP1 to support viral replication and/or to inhibit stress granules formation[42]. Of note, the herpesviruses and alphaviruses have been shown to recruit G3BPs by SLiMs having resemblance to the sequence in N[28,43–45]. However, a deeper mechanistic understanding for how viral proteins affect G3BP biology is missing. Given that the N-G3BP interaction was important for SARS-CoV-2 infection and presents a novel antiviral strategy we investigated this interaction in more detail. We first confirmed that the interaction between N and G3BP1 takes place in SARS-CoV-2 infected cells (Fig. 2g). We also confirmed the binding of recombinant full-length N protein to G3BP1 using FP, which revealed an affinity similar to the N peptide (Fig. 2h). To confirm the SLiM mediated interaction in cells, we compared the interactome of N wild type (N wt) to an N protein where we mutated two amino acids in the ΦxFG motif (N 2 A) using label-free quantitative mass spectrometry. This confirmed a highly specific N-G3BP1/2 interaction fully dependent on an intact ΦxFG motif (Fig. 2i, Supplementary Fig. 3c and Supplementary Data 4). Using a similar approach, we quantitatively compared the interactomes of the N protein from MERS-CoV with that of N from SARS-CoV-2, revealing specific binding of G3BPs to N from SARS-CoV-2 (Fig. 2j and Supplementary Data 4). This is in line with our observation that the N peptides containing the ΦxFG motif from SARS-CoV-2 and SARS-CoV bind to G3BPs with high affinity, while the corresponding MERS-CoV peptide bound weakly ($K_D = 2.8 \mu M$ vs. $K_D = 26 \mu M$, Fig. 1b). Similar results were obtained by Western blot, which also showed that the N

protein from HKU1-CoV did not bind G3BPs, consistent with the ProP-PD results (Supplementary Fig. 2 and Supplementary Fig. 3e). Consistently, the ΦxFG motif resulted in the specific co-localization of mCherry tagged N protein from SARS-CoV and SARS-CoV-2 to arsenite-induced stress granules in cells expressing YFP-tagged G3BP1 (Fig. 3a).

**The N ΦxFG motif affects stress granule formation**. Recent publications have reported that SARS-CoV-2 N induces stress granule disassembly[46,47] but the mechanistic basis of this is unclear. To investigate the effect of the SARS-CoV-2 N ΦxFG motif on endogenous stress granule formation we overexpressed YFP-tagged N wt or N 2 A in HeLa cells and stained for endogenous G3BP1 following arsenite treatment. Quantifying the intensity of cytoplasmic G3BP1 foci in cells positive for YFP revealed that N WT expression disrupted stress granule formation more efficiently when the G3BP-binding motif was intact (Fig. 3b, c). Thus, the N-terminal ΦxFG motif of the N protein constitutes the main determinant of G3BP-binding and stress granule disassembly. We next analyzed G3BP1 foci formation and cellular localization of viral dsRNA in relation to N protein expression levels in VeroE6 cells after six hours of SARS-CoV-2 infection (Fig. 3d, e). At this timepoint, a mixture of early and later stage infected cells is observed. In mock-treated cells, we detected no cells with more than two G3BP1 foci and based on this we set the background threshold at three G3BP1 foci per cell (Supplementary Fig. 3f). In infected cells with low levels of N protein (below 10,000 fluorescent units) a large proportion of cells had multiple G3BP1 foci (Fig. 3e). In cells with low levels of N, this protein and viral dsRNA co-localized with G3BP1 to stress granules (Fig. 3d). However, in cells with high levels of N, protein only two out of 11 cells had G3BP1 foci above threshold levels. Collectively our results suggest that low levels of N protein are insufficient to disrupt stress granule formation and instead N and viral dsRNA co-localize with G3BP1 in these structures. Once N concentrations reach a certain threshold, this disrupts stress granules, and this depends on the ΦxFG motif. A possible interpretation of these observations is that SARS-CoV-2 takes advantage of the stress granule RNA machinery during the earlier stages of infection. Consistently, dsRNA and N co-localize with G3BP1 foci and when the N-G3BP interaction is inhibited a reduction of viral replication is observed (Figs. 2f and 3d).

**N rewires the G3BP1/2 interactome through the ΦxFG motif**. To understand how N could affect stress granule formation and G3BP function through the ΦxFG motif we set out to identify cellular G3BP interactors with similar binding motifs. To this end, we screened a novel ProP-PD library that displays the

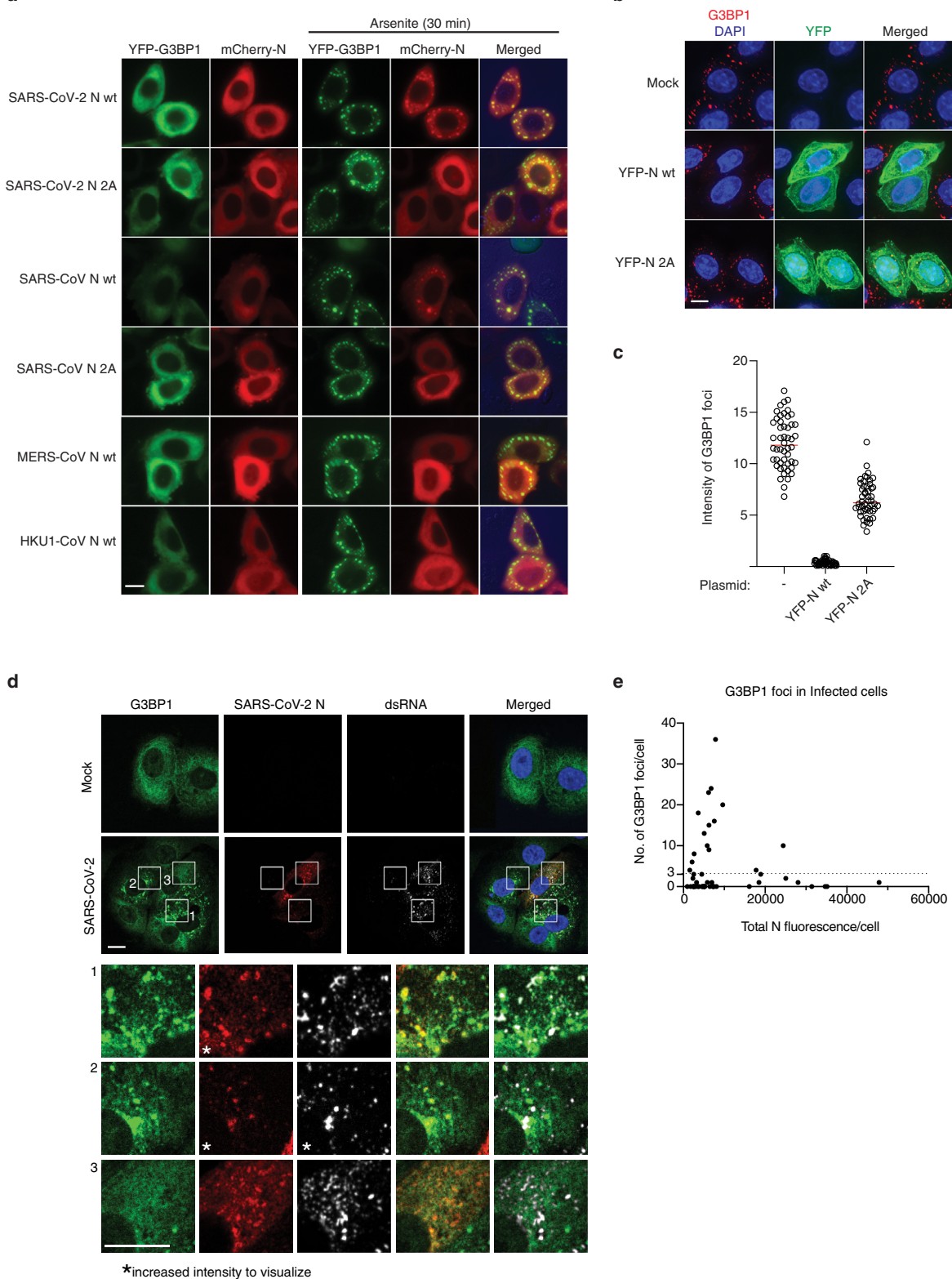

**Fig. 3 Interaction between SARS-CoV-2 N and G3BP1/2 affects stress granule formation. a** Live cell microscopy analysis of HeLa cells co-transfected with YFP-G3BP1 and mCherry tagged N proteins from the viral strains indicated. **b** Effect of SARS-CoV-2 N wt and N 2 A on arsenite-induced stress granule formation as measured by immunofluorescence of endogenous G3BP1. **c** Quantification of G3BP1 foci intensity from **b**. Red bar indicates median intensity, and each circle represents the intensity of one G3BP1 foci. At least five foci from 10 cells were measured. **d** Immunofluorescence analysis of G3BP1, N and viral dsRNA in SARS-CoV-2 infected VeroE6 cells 6 h postinfection. **e** Effect of N levels on G3BP1 foci formation in SARS-CoV-2 infected cells. Each circle represents one cell analysed automatically from one experiment done in duplicate ($n = 51$). Microscopy images shown are representatives of three independent experiments for **a** and **b**. Scale bars are 10 µM. Source data are provided as a Source Data file.

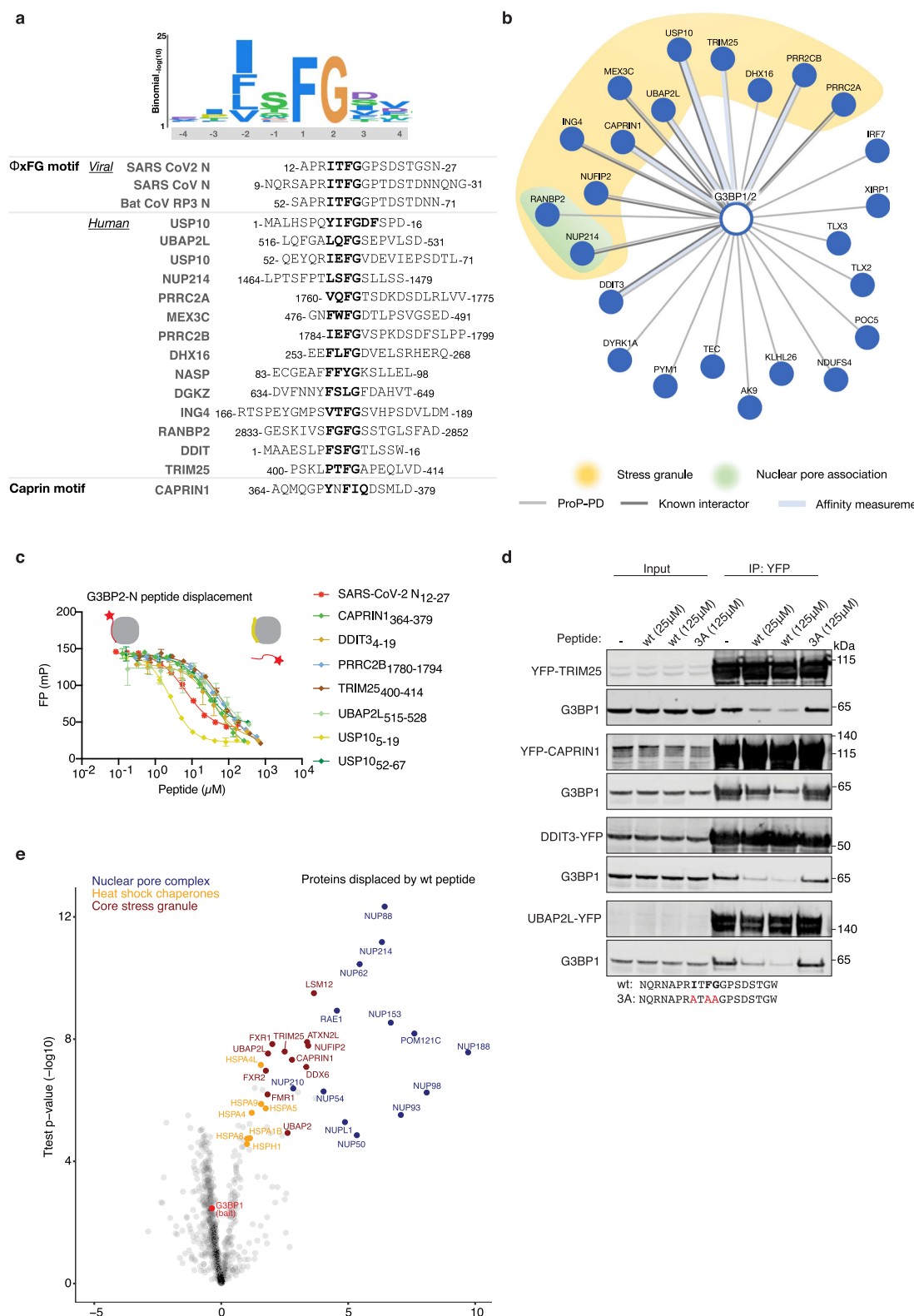

intrinsically disordered regions of the human proteome[22] against the NTF2 domains from G3BP1 and G3BP2. The combined data set includes 72 peptides from 57 proteins with the majority of sequences containing a ΦxFG motif (Φx[FILV]), thus resembling the sequence in the N protein (Fig. 4a and Supplementary Data 5). Nineteen of the proteins uncovered by the screen are in core stress granule proteins, including known peptide motifs in

USP10 and UBAP2L, but also peptides from stress granule proteins that have not previously been reported to contain ΦxFG motifs (Fig. 4a, b). The screen also uncovered a peptide from Caprin-1, which has been shown to bind G3BPs but does not match the consensus sequence[48] (Fig. 4a). This suggests that G3BPs serve as major hubs for stress granule biology in part by interacting with ΦxFG like motifs residing in several stress

**Fig. 4 N competes with cellular proteins for binding to G3BP1/2 through its ΦxFG motif. a** Schematic of the position-specific scoring matrix ΦxFG for G3BP NTF2 domains and sequence alignment of coronavirus peptides and selected human G3BPs ligands found through ProP-PD. **b** Network of a select set of human SLiM-based interactions of the G3BPs found through ProP-PD. Light grey line indicates interactions validated by affinity measurements (Supplementary Data 3), the weight of the line represents the affinity of the interaction (thick, 1–10 μM; medium, 11–100 μM; thin, 101–500 μM). Dark grey lines indicate protein-protein interactions with additional reported evidence. **c** The SARS-CoV-2 ΦxFG N peptide competes with the indicated peptides for G3BP2 binding in vitro (n= two biological duplicates each containing three technical replicates). Shown is a representative plot. Error bars expressed as mean ± SD. **d** The N ΦxFG peptide but not a control peptide competes with TRIM25, CAPRIN1, DDIT2 and UBAP2L for binding to G3BP1 in cells. Shown is a representative blot from three independent experiments. **e** G3BP1-YFP was affinity purified from HeLa cells in the presence of either a SARS-CoV-2 ΦxFG N wt or N 3 A peptide and analysed by quantitative mass spectrometry. The volcano plot shows the two-sided unpaired T test results for all quantified proteins based on four technical replicates per condition.

granule components. However, the screen also returned many peptides in proteins with roles outside of stress granule biology, such as TRIM25 and IRF7 (antiviral interferon signaling)[42], and DDIT3 (endoplasmic reticulum stress)[49]. FP measurements were used to confirm binding between the purified NTF2 domain of G3BP2 and several identified peptides originating from TRIM25, DDIT3, UBAP2L, Caprin-1, USP10, and PRRC2B (Fig. 4c and Supplementary Fig. 4a). Furthermore, we biochemically validated a number of the G3BP-binding motifs in the context of the full-length proteins (Supplementary Fig. 4b). Caprin-1 and UBAP2L co-localized with G3BP1 in stress granules after arsenite treatment in a manner dependent on intact SLiMs (Supplementary Fig. 4c). Conversely, no stress granule localization was observed for TRIM25 and DDIT3 (Supplementary Fig. 4c) further supporting the notion that the G3BPs also have cellular roles beyond stress granule biology[50].

The N protein is a highly expressed viral protein[51] during infection so we hypothesized that it would compete with host cell proteins containing ΦxFG SLiMs for binding to G3BPs. Consistently, FP measurements revealed competition between the N ΦxFG peptide and all of the 7 peptides we tested for interaction with G3BP2 (Fig. 4c). Next, immunopurifications of full-length YFP-tagged TRIM25, DDIT3, Caprin-1, and UBAP2L in the presence of either a N wt peptide or a N 3 A peptide where the ΦxFG motif is mutated to AxAA were performed. As expected, the N wt peptide disrupted interactions to G3BP1 thus validating a direct competition between the N ΦxFG peptide motif and four G3BP-binding proteins (Fig. 4d). The observed competition between the viral N ΦxFG peptide and UBAP2L for binding to G3BP1 is particularly interesting since UBAP2L is required for stress granule assembly through a direct interaction to the G3BPs via its ΦxFG like motif[27,29]. This suggests a mechanistic basis for the ability of the N protein to inhibit stress granule formation.

Given the high levels of N during infection, we speculated that it could mediate a general rewiring of the G3BP interactome through its ΦxFG motif. To test this on a global scale, we purified G3BP1-YFP from HeLa cells and added either N wt or the N 3 A mutated peptide as competitors for cellular proteins. Quantitative label-free mass spectrometry allowed us to determine the proteins being specifically displaced by the N wt peptide (Fig. 4e and Supplementary Data 4). This revealed specific displacement of 59 proteins, including several core stress granule components. In addition, the N peptide also displaced a large number of nuclear pore complex components, heat shock chaperones of the Hsp70 family and proteins of the ASC-1 and CTLH complexes. Except the CTLH components, all of these proteins have been reported to localize to stress granules[27,52,53]. The displacement of nucleoporins from G3BP1 by the N peptide suggests that FG motifs, which are abundant in nucleoporins[54] might recruit them to stress granules through direct interaction to the NTF2 domain of the G3BPs. Consistently, two ΦxFG like motifs from nucleoporins were selected in the G3BP ProP-PD screen (Fig. 4a, b and

Supplementary Data 5). Importantly, Nup62, which we identify in our MS competition screen (Fig. 4e) has been shown to be required for efficient SARS-CoV-2 infection[3]. It is possible that N displaces Nup62 from G3BPs to make it accessible for other viral processes. Together, we show that the N protein modulates the G3BP1/2 host interactome through its ΦxFG motif by competing with numerous cellular ΦxFG containing proteins. Our G3BP motif and mass spectrometric screens provide a rich resource for the future dissection of basic stress granule biology and G3BP signaling in general.

## Discussion

Collectively, we describe a potentially therapeutic relevant interaction between the ΦxFG SLiM in SARS-CoV-2 N and the G3BP proteins. Our results reveal that the N protein during infection hijacks G3BPs to viral replication centers likely to facilitate replication and possibly other aspects of viral RNA metabolism. The disruption of stress granules at later stages of infection could also dampen the cellular antiviral response. Consistent with this idea we identify ΦxFG motifs in TRIM25, MEX3C and IRF7 that are key components of the G3BP-RIG-1 antiviral interferon pathway[42,55–57].

By screening the intrinsically disordered regions of 229 RNA viruses against a host factor in one we uncovered both common principles shared by several viruses as well as interactions specific for a given strain. We show that the SLiMs can be screened for antiviral activity to pinpoint therapeutically relevant interactions. Given the high-resolution binding site information provided by the ProP-PD, this can guide the development of agents targeting these interactions. Peptide-based inhibitors are highly specific but a challenge for peptides as compared to small molecules is their poor pharmacokinetic properties, which may be improved through peptide modifications[58–60]. The clinical use of the HIV peptide inhibitor enfuvirtide (T20) demonstrate that the use of peptides represents a viable strategy for the development of therapeutic antiviral agents[61], although the T20 peptide does not need to cross the cell membrane to elicit an effect. As far as we know, no cell-permeable peptide is used in the clinic, but there are several ongoing clinical trials[62]. The ligands described here may thus represent potential starting points for the development of antiviral agents.

Our approach is easily applied to other relevant host factors and the library can be readily updated to incorporate novel RNA viruses emerging in the future. In this work, 139 recombinant host protein baits were produced for the RiboVD library screen. Still, this set of baits only represents a small fraction of host protein domains expected to recognize virally encoded SLiMs. Thus, the production of recombinant protein baits in considerable numbers and of sufficient purity constitute a bottleneck of the ProP-PD approach. Nevertheless, we foresee that this approach can be a powerful tool for future investigations of virus interactions with cellular host factors and for developing novel antivirals.

## Methods

**RiboVD library design—defining the RNA virus search space.** The RiboVD library was designed using a previously described design pipeline[22]. Following this pipeline, the first step was to define the RNA virus search space. The search space was defined as the UniProt reference proteomes of the mammalian and avian RNA viruses (Riboviria; taxonomic identifier: 2559587) and a representative proteome from RNA virus clades without a reference proteome (A complete list of the viral strains in the library is available at http://slim.icr.ac.uk/phage_libraries/rna_viruses/species.html). This set of 229 viral strains was analysed to define intracellular and intrinsically disordered protein regions. First, the UniProt defined transmembrane and the extracellular regions of transmembrane proteins were removed. Next, the intrinsically disordered region of the remaining search space was defined using 3 approaches: (i) Surface Accessibility Scores from structures of the protein; (ii) Surface Accessibility Scores from homology mapped structures; and (iii) Disorder Predictions using IUPred[63].

**RiboVD library design—library peptide definition.** The intracellular and intrinsically disordered protein regions of the RNA viruses were processed to produce a tiled peptide library. A 16 residue window was scanned across the proteins of the search space until it reached a peptide where at least eight of the 16 amino acids are defined as occurring in an intrinsically disordered region based on the rules above. The window was then shifted forward by four amino acids to produce a library of 16 amino acid peptides overlapping by 12 amino acids. Cytoplasmic loops of length eight or greater that are predicted as disordered were retained. In addition, to avoid issues with unpaired cysteines on the phage coat, all cysteines were replaced with alanines. The resulting library contained 19,549 unique peptides from 1074 proteins (i.e., the numbers are based uncleaved polyprotein) in 229 viral strains across 211 families. The library design contains 4172 coronavirus peptides mapping to 192 proteins from 23 strains including SARS-CoV-2, SARS-CoV, and MERS-CoV-2. The details of the library designs including the viral strains, proteins, peptides and statistics are available at http://slim.icr.ac.uk/phage_libraries/rna_viruses/species.html.

**RiboVD library design—library oligonucleotide definition.** Each peptide from the design was reverse translated into oligonucleotides optimising for codon usage for *E. coli* by stochastically choosing codons to match codon usage in *E. coli*. In the cases where no overlapping peptide existed for a peptide in the design, a peptide was at the N-terminus or a peptide at the C-terminus: two distinct oligonucleotides encoding the same peptide were added. Next, primers for annealing to the phagemid (5' CAGCCTCTTCATCTGGC and 3' GGTGGAGGATCCGGAG) were added to each end of the oligonucleotide design. Finally, oligonucleotides were checked for SmaI restriction sites (GGGCCC or CCCGGG) or self-complementarity of greater than seven contiguous nucleotides and redesigned if necessary. The final library included 23,271 oligonucleotides encoding 19,549 unique peptides in 1074 proteins/polyprotein from 229 viruses 211 in viral families.

**RiboVD phage library construction.** The designed oligonucleotide library was obtained from a commercial provider (Genscript) and used to construct the RiboVD phage library that display the encoded peptides on the major coat protein P8 of the M13 phage following a published protocol[20,22]. In brief, the oligonucleotide library was used as the template for 15 cycles amplification cycles of PCR amplification (98 °C for 10 sec, 56 °C for 15 sec, and 72 °C for 10 sec) using Phusion polymerase (Thermo Scientific) and primers complementary to the constant annealing regions flanking the designed library sequences. The PCR product was purified using a nucleotide removal kit (Qiagen), phosphorylated using T4 polynucleotide kinase (Thermo Scientific) for 1 h at 37°C and annealed to phagemid ssDNA (90 °C for 3 min, 50 °C for 3 min, and 20 °C for 5 min). dsDNA was synthesized using T7 DNA polymerase (Thermo Scientific) and T4 DNA ligase (Thermo Scientific) at 20°C for 16 h. The dsDNA was purified from a 1% agarose gel and electroporated into *E.coli* SS320 (Lucigen) electrocompetent cells pre-infected with M13KO7 helper-phage (ThermoFisher). The phages were allowed to propagate for 24 h in 2YT (10 g yeast extract, 16 g tryptone, 5 g NaCl per L) medium. The phage was precipitated from the supernatant by the addition of 1/5th volume of 20% PEG8000/2.5 M NaCl and centrifugation at $27,000 \times g$ for 20 min. The phage pellets were dissolved in phosphate-buffered saline (PBS, 137 mM NaCl, 2.7 mM KCl, 95 mM $Na_2HPO_4$, 15 mM $KH_2PO_4$ pH 7.5). The resulting phage library was re-amplified and stored at −80°C in the presence of 10% glycerol.

**Protein purification.** *E.coli* BL21-Gold(DE3) (Agilent Technology) transformed with the plasmids encoding the His-GST-tagged proteins (Supplementary Data 1) were grown in 500 mL 2YT at 37°C until an $OD_{600}$ between 0.6 and 0.8. Protein expression was induced with 1 mM isopropyl ß-D-1-thiogalactopyranoside (IPTG) and allowed to proceed for 20 h at 18 °C. The bacteria were harvested by centrifugation ($4500 \times g$, 10 min). Proteins were batch purified using Glutathione Sepharose™ 4 Fast Flow Media (Cytiva), or $Ni^{2+}$ IMAC (Ni Sepharose™ 6 Fast Flow, Cytiva) following the manufacturer's protocols. The purity of the proteins was validated via SDS-PAGE.

Proteins used for FP affinity measurements were expressed and purified in larger-scale bacterial cultures (at least 8 × 500 mL 2YT). His-GST-tagged G3BP1 and G3BP2 were used directly for affinity measurements after dialysis. For the other proteins, the tag was removed by cleaving with HRV 3 C protease or thrombin, and further purified by reverse IMAC or by using a HiTrap Benzamidine FF (HS) (Cytiva) before being dialysed to a suitable buffer for FP affinity measurements (50 mM phosphate buffer, pH 7.5, and 1–2 mM reducing agent (DTT or TCEP)), except for the FERM domains of Radixin and Ezrin (20 mM HEPES pH 7.3, 150 mM NaCl, 3 mM DTT).

**ProP-PD selections and NGS data analysis.** Phage display selections against the immobilized bait proteins were performed in triplicate selections for four rounds of selection. The RiboVD library was used in selections against 139 human bait proteins purified (Supplementary data 1). The GST-tagged NTF2 domains of G3BP1 and G3BP2 were further used in selections against a phage library that encodes the intrinsically disordered regions of the human proteome (the second generation human disorderome displayed on the major coat protein P8 (HD2 P8))[22]. GST-tagged bait protein or GST (10 μg in 100 μL PBS) was immobilized in 96-well Flat-bottom Immunosorp MaxiSorp plates (Nunc, Roskilde, Denmark) for 18 h at 4°C. Wells were blocked with 200 μL BSA (0.5% in PBS) and then washed 4 times with 200 μL PT (PBS + 0.05% (v/v) Tween20). The phage library ($10^{11}$ phage in 100 μL PBS per well) was transferred to the GST-coated wells. After 1 h the library was transferred to bait protein-coated wells. After 2 h, unbound phage were removed and the wells were washed with 5 ×200 μL PT. Bound phage were eluted with 100 μL log phase E.coli OmiMAX (30 min at 37°C), and the bacteria were then hyperinfected with $10^9$ M13KO7 helper-phage per well (45 min at 37° C). The hyperinfected bacteria (100 μL) were transferred into 1 mL 2xYT supplemented with 100 μg carbenicillin, 30 μg kanamycin, 0.3 mM IPTG and grown over night at 37°C for 18 h. Bacteria were pelleted by centrifugation ($2000 \times g$ for 10 min), and phage supernatants were transferred into a fresh 96-well plate and pH adjusted using 1/10 volume 10x PBS. The remaining bacteria were heat-inactivated for 10 min at 65°C. The resulting phage pools were used as in-phage for the next day of selection.

The peptide-coding regions of the naive RiboVD phage library prior to any selection and the binding enriched phage pools (5 μL) were PCR amplified and barcoded using Phusion High-Fidelity polymerase (Thermo Scientific) for 22 cycles, using a dual barcoding strategy[64]. PCR products were confirmed through 2% agarose gel electrophoresis stained with GelRed, using a 50 bp marker (BioRad). The amount of the PCR products were normalized using Mag-bind Total Pure NGS (Omega Bio-tek) before pooling the samples. The resulting amplicon pool was further purified from a 2% agarose gel (QIAquick Gel extraction Kit Qiagen) with GelRed staining and eluted in TE (10 mM Tris-HCl, 1 mM EDTA. pH 7.5). The sample was analyzed using Illumina MiSeq v3 run, 1x150bp read setup, 20% PhiX by the NGS-NGI SciLifeLab facility. Results were processed using in-house custom Python scripts described previously[22]. Reads with an average quality score of 20 or more were kept, and their adapter regions and barcodes were determined allowing a maximum of 1 mismatch per adapter and/or barcode. The unambiguously identified reads were demultiplexed and their adapter and barcode regions were trimmed. Reads from each selection experiment were then translated into amino acid sequences and the number of counts for each unique peptide was determined.

**Peptide mapping and annotation.** An RNA virus search database was added to the PepTools (http://slim.icr.ac.uk/tools/peptools/) web server[22], a tool developed for the annotation of protein regions built on the framework of the PSSMSearch tool[65]. The selected peptide sequences were mapped to the viral and human proteomes with PepTools and annotated with information of the bait and prey (known interaction or shared localisation and functional terms). To assess the quality of RiboVD library we analyzed the coverage percentage of the phage library over the library design using the NGS results of non-challenged naive library phage aliquots. 95.5% of the peptide sequences designed to be in the library were confirmed by the NGS analysis of the naive phage library.

**Analysis of the ProP-PD selection data.** Following our recently outlined ProP-PD data analysis approach[22] four metrics were used to rank the peptides i) NGS read counts, ii) peptide occurrence in replicated selections, iii) number of overlapped peptides, iv) motif match. The four metrics were then combined into a single score called 'Confidence level' forming 3 categories: high (4 metric criteria matched); medium (2 to 3 criteria matched); and low (only 1 metric is matched). Due to the relatively small size of the RiboVD dataset, the coronavirus dataset was further filtered for target-specific ligands (occurring in less than 10 unrelated selections) (Supplementary data 2). The G3BPs HD2 P8 data were combined and a joint confidence score was calculated (Supplementary data 5). The peptide data were combined with available information on stress granule localized proteins from mass spectroscopy of purified mammalian stress granules[27,52,66] and from other studies based on the information listed in the HIPPIE database[67]. Each ProP-PD selected peptide was annotated with the above data and a count of the number of sources of evidence of stress granule localisation.

**PPI network visualization.** Network's visualization was done using Cytoscape[68] using the data provided in Supplementary data 2 and 5.

**FP affinity determinations**. FP experiments were carried out using a SpectraMax iD5 multimode microplate reader (Molecular Devices) using a black half area 96-well plate (Corning, USA #3993) with a total volume of 50 μL. The settings used were 485 nm for excitation and 535 nm for emission.

Peptides were from GeneCust (France) with a purity of over 95%. FITC-labeled peptides were dissolved in dimethyl sulfoxide (DMSO). Unlabelled peptides were dissolved in phosphate buffer (50 mM phosphate buffer pH 7.5 and 1–2 mM reducing agent DTT or TCEP). All measurements were performed at least in triplicates. For saturation binding experiment, proteins were serially diluted in the plate with phosphate buffer in a volume 25 μL, and then supplemented with a master mix of 10 nM FITC-labeled peptide in phosphate buffer. In the displacement assay, a master mix was prepared containing 10 nM FITC-labeled peptide with the protein of interest at a concentration 4 x the $K_D$ determined through direct binding. Twenty-five mitroliter of this displacement master mix was added to unlabeled peptides, which were serially diluted in the plate in 25 μL phosphate buffer.

The data were analysed with GraphPad Prism version 9.0.0 for MacOS (GraphPad Software, San Diego, California USA, www.graphpad.com). The saturation binding data were fitted to a quadratic Eq. (1):

$$Y = A * \frac{pept + X + K_D - \sqrt{(pept + X + K_D)^2 - 4 * pept * X}}{2} + B \quad (1)$$

in which Y is the observed signal, $pept$ is the constant concentration of the FITC-labeled probe peptide, X is the varying protein concentration, A is the signal amplitude divided by the peptide concentration and B is the plateau value obtained for the unbound probe peptide.

The data of the displacement experiment were fitted to a sigmoidal dose-response (2):

$$Y = Bottom + \frac{Top - Bottom}{1 + 10^{(\log IC50 - X) * HillSlope}} \quad (2)$$

in which Y is the observed signal, X is the log of the variable concentration of the unlabeled peptide, and the Top and Bottom are the plateau values obtained for the protein-bound FITC-labeled probe peptide (Top) and the unbound probe peptide (Bottom).

**Cell culture, virus, and reagents**. HeLa and HEK293 cells were maintained in DMEM GlutaMAX containing 100 U/mL penicillin, 100 mg/ml streptomycin, and 10% FCS (all from Thermo Fisher Scientific). Stable HEK cell lines were generated using the T-Rex doxycycline inducible Flp-In system (Invitrogen) and cultivated like HeLa cells with the addition of 5 μg/mL Blasticidin and 100 μg/mL Hygromycin B. Caco2 cells were kept in MEM (Gibco™, 11095-080) supplemented with 20% FBS, penicillin-streptomycin solution (Sigma, P4333), 1 mM sodium pyruvate (ThermoScientific™, 11360039), and nonEssential amino acids (Gibco™, 11140035).

VeroE6 cells were cultured in DMEM (Sigma) supplemented with 5% fetal bovine serum (FBS), 100 U/mL of penicillin and 100 μg/mL streptomycin (Gibco). The patient isolate SARS-CoV-2/01/human/2020/SWE accession no/GeneBank no MT093571.1, was provided by the Public Health Agency of Sweden. SARS-CoV-2 passage number 4 was cultured and titrated in VeroE6 cells. E. coli DH5α were maintained and propagated using standard microbiological procedures. The following drug concentrations were used: sodium arsenite 0.5 mM, doxycycline 10 ng/ml unless otherwise stated.

**Expression constructs and cell line generation**. Standard cloning techniques were used throughout. All N proteins and variants were generated by gene syntesis (Geneart). To generate the YFP-G3BPi inhibitor, the double FGDF motif from Semliki Forest virus nsP3 (RTTFRNKLPFTFGDFDEHEVDALASGITFGDFDDVL) or a control inhibitor (RTTFRNKLPATAGDFDEHEVDALASGITAGDADDVL) were fused to the C terminus of YFP by cloning the DNA encoding this into the pcDNA5/FRT/TO vector (Invitrogen). DNA encoding the G3BPi sequences was purchased from GeneArt, Life Technologies. All constructs were fully sequenced. The following point mutations were introduced using quick-change mutagenesis to uncouple binding to G3BP: TRIM25 (F406A G407A), DDIT3 (F10A G11A), CAPRIN-1 (Y370S N371K F372S I373T), UBAP2L (F518L F523G). See supplementary primer table for names and sequences of primers used in this study. Detailed mutagenesis and cloning strategies are available upon request.

**Lentivirus production, transductions, and virus infection**. Transfer plasmids for lentiviral transduction were ordered from GenScript. To generate transfer plasmids, four copies of inhibitory peptide (three copies for the Semilikiforest peptide) or control peptides with the binding motifs mutated were fused to C-terminus of EGFP and cloned to pLJM1-EGFP vector (David Sabatini lab, Addgene plasmid #19319).

Lenti-X 293 T cells (Takara bio) grown in a 100 mm plate format were co-transfected with 4.5 μg psPAX2 (Didier Trono lab, Addgene plasmid #12260), 500 ng pMD2.G (Didier Trono lab, Addgene plasmid #12259) and 5 μg of transfer plasmid per plate using polyethylenimine (Merck) as transfection reagent. At 24 h post-transfection, the medium was replaced with DMEM GlutaMAX containing 100 U/ml penicillin, 100 mg/ml streptomycin, and 10% FCS. Viral supernatant was

harvested at 72 h post-transfection, filtered through a 0.22 μm low protein-binding syringe filter and frozen at −80 °C.

For transductions, VeroE6 or HEK cells were seeded into greiner CELLSTAR® 96-well plates or 6-well plates (VWR) containing lentivirus in DMEM containing 2% FBS and 1 μg/mL polybrene, and incubated for 72 h. Transduced VeroE6 cells were infected with SARS-CoV-2 using the indicated multiplicity of infection (MOI). Virus was detected using the same method as for viral titration except for using donkey antirabbit IgG Alexa Fluor 555 secondary antibody (Invitrogen, A32794). Nuclei were counterstained by DAPI. The number of infected cells were determined by quantifying cells positive for SARS-CoV-2 nucleocapsid using a TROPHOS Plate RUNNER HD®. The number of infected cells were determined by dividing the SARS-CoV-2 nucleocapsid count by the DAPI count and presented as percentage infection of in relation to cells expressing control peptide.

**Viral titration**. The virus was diluted in ten-fold dilutions and added to VeroE6 cells followed by 1 h incubation at 37°C and 5% CO2. After 24 h of infection, the inoculum was removed and cells were fixed in 4% formaldehyde for 30 min, permeabilized in PBS 0.5% trition-X-100 and 20 mM glycine. The virus was detected using primary monoclonal rabbit antibodies directed against SARS-CoV-2 nucleocapsid (Sino Biological Inc., 40143-R001), and secondary antirabbit HRP conjugated antibodies (1:2000, Thermo Fisher Scientific). Viral foci were then stained by incubation with TrueBlue peroxidase substrate for 30 min (KPL, Gaithersburg, MD).

**Antibodies**. The following antibodies were used at the indicated dilutions: c-Myc (1:1000, Santa Cruz Biotechnology, sc-40), rabbit anti-G3BP1 (WB; 1:1000, Cell Signaling Technology, #17798 S), mouse anti-G3BP1 (IF; 1:1000, Abcam, ab56574), GFP-Booster_Atto488 (IF 1:300, ChromoTek), mouse anti-GFP (WB, 1:1000, Roche, 11814460001), rabbit anti-GFP (WB; 1:5000, in-house), rabbit anti-SARS-CoV-2 nucleocapsid(WB 1:2500; IF 1:500; Sino Biological Inc., 40143-R001), mouse APC-conjugated antibody directed against dsRNA J2 (IF 1:200, Scicons, 10010500) mouse anti-3xFlag M2 (WB 1:25000, Sigma, F1804), rabbit anti-Tubulin (WB 1:4000, Abcam, ab6046), antirabbit SARS-CoV-2 nucleoprotein (1:500; Invitrogen, MA5-29981), antimouse G3BP1 (1:1000; SantaCruz Biotechnology, sc-365338), antimouse GAPDH (1:1000; SantaCruz Biotechnology, sc-47724), donkey antirabbit IgG (Invitrogen, A32794), goat antirabbit HRP conjugated antibody (WB 1:2000 or 1:5000, Thermo Fisher Scientific, 31460), goat antimouse HRP conjugated antibody (WB 1:5000, Thermo Fisher Scientific, 31430), goat antimouse Alexa Fluor 546 (IF 1:1000, Invitrogen, A-11003), donkey antirabbit Alexa555 (IF 1:500, Thermo Fisher Scientific, a31572), donkey antimouse Alexa488 (IF 1:500, Thermo Fisher Scientific, a21202), goat IRDye 800CW anti-Mouse IgG (WB 1:10000, Li-Cor Biosciences, 926-32210), goat IRDye 800CW anti-Rabbit IgG (WB 1:10000, Li-Cor Biosciences, 926-32211), goat IRDye 680RD anti-Mouse IgG (WB 1:10000, Li-Cor Biosciences, 926-68070), goat IRDye 680RD anti-Rabbit IgG (WB 1:10000, Li-Cor Biosciences, 926-68071).

**RNA isolation, cDNA synthesis, and qPCR**. Total RNA was isolated from cells and 400 ng was used to synthesize cDNA using High Capacity cDNA Reverse Transcription Kit (Applied Biosystems) according to the manufacturer's instructions. GAPDH transcripts were detected by RT2 qPCR Primer Assay (Qiagen, Cat# 330001 PPQ00249A) and the qPCRBIO SyGreen Mix Hi-ROX kit (PCRBIOSYSTEMS), SARS-CoV-2 transcripts were detected using forward (GTCATGTGTGGCGGTTC ACT) and reverse (CAACACTATTAGCATAAGCAGTTGT) primers and probe (FAM-CAGGTGGAACCTCATCAGGAGATGC-BHQ) and the qPCRBIO Probe Mix Hi-ROX kit (PCRBIOSYSTEMS). qPCR was run using a StepOnePlus fast real-time PCR system (Applied Biosystems).

**Live cell imaging**. Live-cell analysis was performed on a Deltavision Elite system using a × 40 oil objective with a numerical aperture of 1.35 (GE Healthcare). The DeltaVision Elite microscope was equipped with a CoolSNAP HQ2 camera (Photometrics). Cells were seeded in eight-well Ibidi dishes (Ibidi) and before filming, the media was changed to Leibovitz's L-15 (Life Technologies). Appropriate channels were recorded for the times indicated. For transient transfections, DNA constructs were transfected into HeLa cells using lipofectamine 2000 (Life Technologies) 24 h prior to analysis.

**Immunoprecipitations**. Cells were lyzed in lysis buffer (150 mM NaCl, 50 mM Tris pH 7.4, 1 mM EDTA, 1 mM DTT, 0.1% NP40) supplemented with protease and phosphatase inhibitors (Roche) for 25 min on ice. Lysates were cleared for 15 min at 20000 x g and incubated with 20 μL preequilibrated GFP-trap or Myc-trap beads (ChromoTek) as indicated for 45 min at 4 °C. Following 3 washes with lysis buffer, the beads were either eluted in 25 μL 2x LDS sample buffer (Novex, Life Technologies), boiled for 5 min, separated by SDS-PAGE and analyzed by Western blotting (LI-COR ImageStudio v. 3.1.4 used for analysis) with the indicated antibodies or subjected to quantitative mass spectrometry as described in the AP-MS section. For peptide competition experiments the indicated peptides were added to cell lysates for 30 min at 4 °C before incubated with GFP-trap beads. For uncropped Western blots, see Supplementary Fig. 5.

**Immunoprecipitation of SARS-CoV/SARS-CoV2/MERS-CoV N proteins**. Two 15 cm³ dishes were seeded with HeLa cells at 20% confluency. On the following day, cells were transfected with 2.5 mg YFP-myc-N protein or YFP control plasmids. Cells were collected after 48 h and lysed in 900 mL lysis buffer: 100 mM NaCl, 50 mM Tris pH7.4, 0,1% IGEPAL (NP40), 1 mM DTT supplemented with protease (Complete EDTA Free mini:Roche) and phosphatase (PhosStop: Roche) inhibitor tablets. Lysates were sonicated with Bioruptor for 10 cycles: 30 s ON, 30 s OFF intervals at 4 °C and cleared at $20000 \times g$ for 45 min. The cleared lysate was incubated with pre-equilibrated GFP-Trap beads for 1 h at 4 °C and rotation. Three washes with 1 mL wash buffer: 150 mM NaCl, 5 0 mM Tris pH 7.4, 0.05% IGEPAL (NP40), 5% glycerol, 1 mM DTT, followed by one wash with 1 mL basic wash buffer: 150 mM NaCl, 50 mM Tris pH 7.4, 5% glycerol. The supernatant was discarded, and beads were stored at −20 °C before prepared for the MS analysis, or separated by SDS-PAGE followed by Western blot analysis.

**G3BP1 immunoprecipitation in SARS-CoV-2 infected cells**. Caco2 cells were seeded in a 6-well plate and incubated 48 h at 37 °C. Cells were infected with SARS-CoV-2 at a MOI of 1 and incubated for 48 h. Cells were, then, lysed in 300 μL of Immunoprecipitation lysis buffer (20 mM HEPES, pH 7.4, 110 mM potassium acetate, 2 mM MgCl₂, 0.1% Tween 20, 1% Triton X-100, 0.5% sodium deoxycholate, 0.5 M NaCl) and sonicated for 2 min at 75% amplitude. The lysate was cleared by centrifugation 10 min at 14000 x g and 40 μL of the lysate was aliquoted for further analysis by SDS-PAGE followed by western blot analysis. The remaining lysate was used for immunoprecipitation studies. Shortly, 250 μL of cell lysate was incubated with 800 ng of G3BP1 antibody (SantaCruz, #sc-365338) for 15 min. The lysate was incubated overnight with 20 μL of protein G Mag Sepharose™ (#28-9613-79; GE Healthcare) and washed with lysis buffer. Proteins were eluted in 40 μL of glycine (pH 2.2) and equilibrated with 10 μL Tris (pH 9.1). NuPAGE LDS sample buffer was added to the lysate and immunoprecipitation samples and samples were boiled for 10 min at 95 °C. Samples were analysed by SDS-PAGE followed by western blotting. For western blot analysis, proteins were transferred to a PVDF membrane (0.45 μm, BioRad). The membrane was blocked 1 h at room temperature in 5% milk. Membranes were incubated for 1 h with primary antibodies: SARS-CoV-2 nucleoprotein (1:500), G3BP1 (1:1000) and GAPDH (1:1000); followed by incubation with secondary antibody: antimouse HRP conjugated (1:30,000). Membranes were incubated with developing reagents and imaged with GelDoc imaging station.

**Immunofluorescence microscopy**. For immunofluorescence microscopy, HeLa cell lines were seeded in eight-well Ibidi dishes (Ibidi) and transfected with the indicated constructs. Twenty-four hours after transfection cells were treated with 0.5 mM sodium arsenite for 30 min to induce the formation of stress granules and subsequently fixed in 4% paraformaldehyde in PBS. Cells were blocked in 3% BSA in PBS-T for 30 min before incubation with GFP-Booster _Atto488 (1:300; Chromotek), or mouse anti-G3BP1 in 3% BSA in PBS-T for 1 h at room temperature. Unbound primary antibodies were removed by washing four times for 5 min in PBS-T at room temperature followed by incubation with secondary antibodies (Alexa Fluor 546; 1:1000; Invitrogen) and DAPI for 45 min. Ibid dishes were then washed four times for 5 min in PBS-T. Z stacks 200 nm apart were recorded on a microscope (DeltaVision Elite) using a 40 × oil objective lens (numerical aperture 1.35) followed by deconvolution using SoftWoRx. The fluorescent intensity of stress granule signals was quantified by drawing a circle closely around stress granule signals and the intensity values from the peak continuous stacks were subtracted from the background of neighboring areas.

VeroE6 cells were seeded in eight-well chamber slides (Sarstedts) and infected with SARS CoV-2 for 6 h. Cells were fixed with 4% formaldehyde, quenched with 10 mM glycine, and permeabilized with PBS and 0.5% Triton X-100. Thereafter, cells are incubated with primary antibodies against SARS-CoV-2 nucleocapsid ((1:500) Sino Biological Inc., 40143-R001) and G3BP1 ((1:500) Abcam, ab56574) followed by incubation with conjugated secondary antibodies antirabbit Alexa555 and antimouse Alexa488 (1:500, Thermo Fisher Scientific). Then cells were stained with an APC-conjugated antibody directed against dsRNA J2 ((1:200) Scicons 10010500, the antibody was conjugated using APC Conjugation Kit - Lightning-Link® (ab201807)). Nuclei were detected with DAPI (diluted 1:1500), coverslips were mounted and samples were analyzed using a Zeiss 710 (Carl Zeiss, Oberkochen, Germany) confocal microscope with a 63x oil objective (Zeiss) and ZEISS ZEN Imaging Software. For quantification of G3BP1 foci and nucleocapsid, images were obtained using a Leica SP8 Laser Scanning Confocal Microscope with a 63x oil objective (Leica) and Leica Application Suit X software (LAS X, Leica). A total of 10 images containing 51 cells from infected samples and 14 cells from mock samples were quantified using ImageJ/Fiji. An area outline was drawn for each cell and the total fluorescent signal of nucleoprotein and amount of stress granules/area was counted using "analyze particles". The threshold was set equal for all measurements and cells with saturated signal was excluded. All data were adjusted for background signal.

**Assembly assay**. pcDNA3.1 expression plasmids coding for the structural proteins of SARS CoV-2, Spike (S D614), Membrane (M) (was a gift from Jeremy Luban (Addgene plasmid #158074 and 158078; http://n2t.net/addgene:158078;

RRID: Addgene_158078)), Envelope 3xFlag (pGBW-m4252867 was a gift from Ginkgo Bioworks & Benjie Chen (Addgene plasmid # 153626; http://n2t.net/addgene:153626; RRID: Addgene_153626) and pcDNA5/FRT/TO Myc SARS CoV2 N and N 2 A was transfected into HEK293T cells using GeneJuice (Novagen, Darmstadt, Germany) following the manufacturer's protocol (12 μg of DNA/sample in total). Cells and Virus-Like Particles (VLPs) were collected 24 h after transfection. Cells were lysed (0.5 M Tris-HCl pH 8, 1 M NaCl, 1% Triton X-100) and supernatant of transfected cells was collected, and concentrated by ultracentrifugation ($100,000 \times g$, 90 min 4 °C, SW41, Beckman Coulter, Brea, CA). The pellet was resuspended in reducing Laemmli SDS-PAGE sample buffer. Proteins were separated with SDS-PAGE and Western blot analysis was performed using antibodies against SARS CoV-2 nucleocapsid, 3xFlag M2 (Sigma, F1804) and tubulin (Abcam, ab6046).

**Affinity purification and mass spectrometry (AP-MS)**. Partial on-bead digestion was used for peptide elution from GFP-Trap Agarose (Chromotek). Briefly, 100 μL of elution buffer (2 M urea; 2 mM DTT; 20 μg/mL trypsin; and 50 mM Tris, pH 7.5) was added and incubated at 37 °C for 30 min. Samples were alkylated with 25 mM CAA and digested overnight at room temperature before the addition of 1% trifluoroacetic acid (TFA) to stop digestion. Peptides were desalted and purified with styrene-divinylbenzene reversed-phase sulfonate (SDB-RPS) StageTips. Briefly, two layers of SDB-RPS were prepared with 100 μL wash buffer (0.2% TFA in H₂O). Peptides were loaded on top and centrifuged for 5 min at $500\times g$, and washed with 150 μL wash buffer. Finally, peptides were eluted with 50 μL elution buffer (80% ACN and 1% ammonia) and vacuum-dried. Dried peptides were dissolved in 2% acetonitrile (ACN) and 0.1% TFA in water and stored at −20 °C.

**LC-MS analysis**. Liquid chromatography-mass spectrometry (LC-MS) analysis was performed with an EASY-nLC-1200 system (Thermo Fisher Scientific) connected to a trapped ion mobility spectrometry quadrupole time-of-flight mass spectrometer (timsTOF Pro, Bruker Daltonik GmbH, Germany) with a nano-electrospray ion source (Captive spray, Bruker Daltonik GmbH). Peptides were loaded on a 50 cm in-house packed HPLC-column (75 μm inner diameter packed with 1.9 μm ReproSilPur C18-AQ silica beads, Dr. Maisch GmbH, Germany). Peptides were separated using a linear gradient from 5-30% buffer B (0.1% formic acid, 80% ACN in LC-MS grade H₂O) in 43 min followed by an increase to 60% buffer B for 7 min, then to 95% buffer B for 5 min and back to 5% buffer B in the final 5 min at 300 nL/min. Buffer A consisted of 0.1% formic acid in LC-MS grade H₂O. The total gradient length was 60 min. We used an in-house made column oven to keep the column temperature constant at 60 °C.

Mass spectrometric analysis was performed essentially as described in Brunner et al.[69] in data-dependent (ddaPASEF) mode. For ddaPASEF, 1 MS1 survey TIMS-MS and 10 PASEF MS/MS scans were acquired per acquisition cycle. Ion accumulation and ramp time in the dual TIMS analyzer was set to 100 ms each and we analyzed the ion mobility range from 1/K0 = 1.6 Vs;cm⁻² to 0.6 Vs;cm⁻². Precursor ions for MS/MS analysis were isolated with a 2 Th window for m/z < 700 and 3 Th for m/z > 700 in a total m/z range of 100-1.700 by synchronizing quadrupole switching events with the precursor elution profile from the TIMS device. The collision energy was lowered linearly as a function of increasing mobility starting from 59 eV at 1/K0 = 1.6 VS;cm⁻² to 20 eV at 1/K0 = 0.6 Vs cm⁻². Singly charged precursor ions were excluded with a polygon filter (out of control, Bruker Daltonik GmbH). Precursors for MS/MS were picked at an intensity threshold of 1.000 arbitrary units (a.u.) and resequenced until reaching a 'target value' of 20.000 a.u taking into account a dynamic exclusion of 40 s elution.

**Data analysis of proteomic raw files**. Mass spectrometric raw files acquired in ddaPASEF mode were analyzed with MaxQuant (version 1.6.7.0)[70,71]. The Uniprot database (2019 release, UP000005640_9606) was searched with a peptide spectral match (PSM) and protein level FDR of 1%. A minimum of seven amino acids was required including N-terminal acetylation and methionine oxidation as variable modifications and cysteine carbamidomethylation as fixed modification. Enzyme specificity was set to trypsin with a maximum of two allowed missed cleavages. The first and main search mass tolerance was set to 70 ppm and 20 ppm, respectively. Peptide identifications by MS/MS were transferred by matching four-dimensional isotope patterns between the runs (MBR) with a 0.7-min retention-time match window and a 0.05 1/K0 ion mobility window. Label-free quantification was performed with the MaxLFQ algorithm[72] and a minimum ratio count of two.

**Bioinformatic analysis of LC-MS data**. Proteomics data analysis was performed with Perseus[73] and within the R environment (https://www.r-project.org/). MaxQuant output tables were filtered for 'Reverse', 'Only identified by site modification', and 'Potential contaminants' before data analysis. Missing values were imputed after stringent data filtering and based on a normal distribution (width = 0.3; downshift = 1.8) prior to statistical testing. For pairwise proteomic comparisons (two-sided unpaired t test), we applied a permutation-based FDR of 5% to correct for multiple hypothesis testing including an $s_0$ value[74] of 0.1.

**Reporting summary**. Further information on research design is available in the Nature Research Reporting Summary linked to this article.

## Data availability

The mass spectrometry proteomics data have been deposited to the ProteomeXchange Consortium via the PRIDE partner repository[75] with the dataset identifier PXD025410 available at: http://proteomecentral.proteomexchange.org/cgi/GetDataset?ID=PXD025410. A complete list of the viral strains in the library is available at http://slim.icr.ac.uk/phage_libraries/rna_viruses/species.html. Transmembrane and extracellular regions of transmembrane proteins were identified using UniProt (https://www.uniprot.org). The details of the library designs including the viral strains, proteins, peptides and statistics are available at http://slim.icr.ac.uk/phage_libraries/rna_viruses/species.html. For peptide mapping and annotation, an RNA virus search database was added to the PepTools (http://slim.icr.ac.uk/tools/peptools/) webserver. The identity of proteins localizing to stress granules was retrieved from the HIPPIE database (http://cbdm-01.zdv.uni-mainz.de/mschaefer/hippie/). Intrinsic disorder predictions were made using IUPred (https://iupred3.elte.hu/). Raw data is provided as source data for Figs. 2a, e, f, 3c, e and Supplementary Figs. 3f and 4c. Source data are provided with this paper.

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

## Acknowledgements

This work was supported by the grants from the Swedish Foundation for Strategic research (Y.I., P.J.: SB16-0039), the Swedish Research Council (Y.I.: 2020-03380; PJ: 2020-04395; A.Ö.: 2017-02438, 2018-05851; G.M.: 2018-03843), the Knut and Alice Wallenberg Foundation (Y.I., P.J., and A.Ö. via Science for Life Laboratory, 2020.0182), the Sygeforsikring Danmark (2020-0176 to J.N., M.M., Y.I., G.M.), and by Cancer Research UK (N.D.: C68484/A28159). F.C. acknowledges the European Commission's Horizon 2020 Research and Innovation Programme (Marie Skłodowska-Curie Individual Fellowship grant agreement no. 846795). We thank the medical faculty Umeå University strategic research resource and the Laboratory for Molecular Infection Medicine Sweden for generous support (A.Ö.), and the Biochemical Imaging Center at Umeå University and the National Microscopy Infrastructure, NMI (VR-RFI 2016-00968) for assistance in microscopy. Sequencing was performed by the SNP&SEQ Technology Platform in Stockholm. The facility is part of the National Genomic Infrastructure (NGI) Sweden and Science for Life Laboratory. The SNP&SEQ Platform is also supported by the Swedish Research Council and the Knut and Alice Wallenberg Foundation. Work at the Novo Nordisk Foundation Center for Protein Research is supported by grant NNF14CC0001.

We thank Prof. Sachdev Sidhu for providing constructs for the expression of 59 proteins, Dr. Andreas Ernst ATG8 domain constructs, and Prof. Carlos Fontes and Renaud Vincentelli for constructs for PDZ domain expression. pGEX Grb2 SH2-SH3 was provided by Bruce Mayer (Addgene plasmid # 46442), GST-Hsp90 N(9-236) by William Sessa (Addgene plasmid # 22481), intersectin I SH3 A domain by Peter McPherson (Addgene plasmid # 47413), pGEX-5X-1-LARP7 by Blerta Xhemalce (Addgene plasmid # 113545), pGEX-2T PTP-1B by Ben Neel (Addgene plasmid # 8602), pGEX-4T-1-RIPK3 by Jaewhan Song (Addgene plasmid # 78827), pGEX6P-1-SF2 by Honglin Chen (Addgene plasmid # 99020), pGEX VAMP7 (1-188) by Thierry Galli (Addgene plasmid # 42315), the SARS-CoV-2 pcDNA 3.1 SARS-CoV-2S D614, and M by Jeremy Luban (Addgene plasmids #158074, #158078), pGBW-m4252867 expressing E 3xFLAG by Ginkgo Bioworks & Benjie Chen (Addgene plasmid #153626). We thank the protein production and characterization platform at NNF CPR for producing full-length SARS-CoV-2 N protein. We thank Josephine K. Duel for help with the preparation of samples for MS analysis.

## Author contributions

T.K. performed analysis of stress granules in HeLa cells and analysis of G3BP motifs and competition studies. C.B., M.A., F.M., E.A., and J.K. purified proteins. C.B and Y.I. performed phage selections. Y.I. and A.S. analyzed PPI data. C.B., F.M., M.A., and J.K. performed affinity measurements and analyzed data. D.H.G. prepared samples for N interactomes and western blot analysis of N purifications. R.L. performed viral infections and microscopy, inhibition assays, E.N. performed assembly assays and microscopy. N.D. designed the RiboVD library, M.A. constructed it. L.S. and N.D. analyzed NGS results. A.M.M. and G.M. analyzed G3BP:N interaction in virus-infected cells. R.T.I. generated lentiviral constructs and particles. J.N. performed cloning of N expression constructs. F.C., A.M., M.M. performed mass spectrometry analysis. N.D., P.J., A.Ö., J.N., and Y.I. conceived the study. Y.I., J.N., A.Ö. supervised the project. J.N. and T.K. drafted initial manuscript and all authors contributed to the final version.

## Funding

## Competing interests

J.N. is on the scientific advisory board for Orion Pharma. The other authors declare no competing interests.
