## [Peer Review File · Nature Communications]

Large scale discovery of coronavirus-host factor protein interaction motifs reveals SARS-CoV-2 specific mechanisms and vulnerabilitiesReviewers' Comments:

Reviewer #1:

Remarks to the Author:

Kruse et al describes the use of a large phage library displaying 16-mer peptides covering accessible regions of RNA virus proteins to identify short linear motif (SLiM) in these that can interact with a collection of 57 different protein interaction domains from 53 human proteins. This study reports 269 putative SLiM-based interactions between 44 human protein domains and 64 viral proteins from 18 coronavirus strains, focusing on those involving SARS-CoV-2, SARS-CoV, and MERS-CoV. Functional follow-up studies are described that address strong binding of an ØxFG motif in the nucleocapsid (N) proteins from SARS-CoV-2 and SARS-CoV to the NTF2 domain of the stress-granule protein G3BP1 and G3BP2. Additional ØxFG-containing host cell proteins binding to G3BP1/2 NTF2 domains are identified from a human peptide library. Further studies on viral and cellular ØxFG motifs suggest a mechanistic model for stress-granule disruption by SARS-CoV-2 and SARS-CoV N proteins, and show that blocking of this process has potential for antiviral development by potentially interfering with SARS-CoV-2 replication.

These studies represent a very large and interesting body of work involving diverse experimental approaches. All work has been carried out in a highly professional manner, and the conclusions of the study are well supported by the results presented. A wealth of new information as well as a strong case for the utility of the proteomic peptide-phage display approach in further studies on virus-host interactions are provided. Thus, this study could be defined a landmark paper in its field even though none of the individual findings reported in it constitute a distinct breakthrough.

This study could obviously be extended into a number of directions, and it reports several observations that would be interesting to follow up in more detail. However, given the already extensive nature of this study, such work should probably be left for the future. On the other hand, the paper has no deficiencies or flaws that would clearly need to be addressed by additional experimentation before publication.

Major comments:

The experimental strategy and rationale of the study should be explained better:

1) A study of 1074 viral proteins from 229 RNA viruses is introduced to the reader, but all hits of the screen involve coronaviruses. Presumably also peptides from other viruses were selected by some of the 139 bait proteins used but were excluded from the current results. If so, this should be better explained. And even if the rest of the hits will be reported in future publications, it would be of interest to know what percentage of all the discovered interactions are covered by the 269 coronavirus SLiM-based interactions now listed in Table S2.

2) As pointed out by the authors the SLiM phage library approach can valuably complement more unbiased discovery approaches, such as large scale mass spectrometry or CRISPR-based screens. However, the 139 human bait proteins used cover only a tiny fraction of all relevant protein interaction domains (PID). This limitation should be discussed, and the current rationale of selecting the PIDs used as baits should be better explained. It is mentioned that 57 were selected because they are present in proteins already reported as SARS-CoV-2 interactors, whereas 82 were selected based on some other logic that is not explained. It would also be of interest to know how many of the 269 SLiM-based hits were selected by these 57 vs. 82 baits.

3) Despite the space restrictions the essential bioinformatic basis and design of the large RiboVD library should be clarified instead of just referring to more extensive information in other publications and databases.

4) It is understandable that due to the large collection of advanced methods used in this study the Materials & methods section has been written in a concise manner. Although the quality of the data looks excellent, the validity of many the experimental approaches, for example related to affinity measurement and virological assays, is therefore difficult to judge. A more detailed version Materials & methods-section, perhaps as a supplementary appendix would therefore be helpful.

Reviewer #2:

Remarks to the Author:

This study reported the use of the proteomic peptide-phage display (ProP-PD) platform to dissect the complex interactions of RNA viruses with host cellular proteins. This methodology is based on the documented interactions of short linear interaction motifs in viral proteins with the globular domains of cellular proteins. Thus, other forms of protein-protein interactions are not supposed to be covered by this method.

The authors have designed a ProP-PD library that covers 1074 viral proteins from 229 RNA viruses, including 19549 unique 16 amino acid long peptides. They went on and tested SARS-CoV-2 proteins against a total of 139 host protein domains that have been previously reported to interact with SARS-CoV-2 (57 domains) or other viruses (82 domains). 269 interactions were identified, 27 were validated by fluorescent polarization (FP) affinity assay.

The strength of this approach is to precisely determine the interacting motifs between the viral and cellular proteins. Since the pre-selected nature of the cellular protein domains, this method does not necessarily reveal novel virus-host interactions.

The authors went further to characterize the interactions of the FG motif in SARS-CoV-2 N protein and G3BP1, and showed the inhibitory effect of this short peptide, when fused with GFP, by more than 3-fold. A stronger binding motif FGDF from SFV NSP3 proteins exhibited a much greater inhibition by competitive binding to G3BP1, suggesting a potential therapeutic value of such inhibitory peptides.

More experiments were performed to understand how N protein modulates stress granule formation and how G3BP1 interacts with other cellular proteins, which led to the identification of new G3BP-interacting peptide motifs and new cellular partner proteins.

A few questions to be answered:

1. One aim of this study is to identify peptide-based antiviral drugs. Please discuss any precedent of using peptide as antivirals and the challenges compared to small molecule drugs.
2. Please discuss and acknowledge the strength and limitation of ProP-PD approach in discovering new virus-host interactions.
3. Fig 2G: the signal of N protein in the IP is fairly weak, which questions the importance of this interaction in SARS-CoV-2 infection and the value as antiviral drugs.
4. Fig 3D, E: how did the authors distinguish early stage from late stage SARS-CoV-2 infected cells after 6 hours of infection? Would it be more rational to examine cells at different time points (for example, 4 hours vs 12 hours) to claim early vs late stage infections? In Fig 3D, visually, all infected cells appear to have G3BP1 granules.

Reviewer #3:

Remarks to the Author:
Article summary:

The authors set out to identify viral short linear interaction motifs (SLiMs) that interact with human proteins, with a focus on coronaviruses. The authors began by generating a phage display library of short peptides identified from the predicted unstructured regions of RNA virus proteins (including but not limited to coronaviruses), and utilized this library for in-vitro binding experiments with 57 domains from known human SARS-CoV-2 interacting proteins as well as 82 other human peptide binding protein domains. Reassuringly, the resulting interaction network included known interactions between coronavirus proteins and host proteins, such as interactions between N protein and G3BP1/2, as well as between Nsp14 and AP2.

The authors focused on the linear motif phi-x-F-G in coronavirus N protein and its interaction with the stress granule associated proteins G3BP1/2. They evaluated peptide-based therapeutic interventions for SARS-CoV-2 infection, theorizing such perturbations would interfere with the binding of this linear motif and G3BP and inhibit infection, and successfully verified this hypothesis. They further performed biochemical and cell biological studies to gain insights into the functional and biochemical consequences of the interaction between N protein and G3BP1/2 through the N protein phi-x-F-G motif.

Applying mutations in the phi-x-F-G motif to N protein, they demonstrate convincingly that the wild type motif is required to interfere with stress granule assembly (Figure 3B). The authors also demonstrate that overexpression of the N protein peptide displaces a number of cellular G3BP interactions, raising interesting questions regarding host network rewiring resulting from the N protein – G3BP interaction.

My overall impressions and suggestions:

Systematic mapping of interactions between viral SLiMs and host proteins is a helpful contribution towards our structural understanding of host-pathogen interactions. As the authors rightly point out, their approach builds upon large-scale interactome studies by providing further detail regarding interaction interfaces. This information is useful for the design of small molecule inhibitors, and the authors prove this point by demonstrating a peptide-based inhibitor of coronavirus infection. The authors' methodology and conclusions are sound and are supported by their data.

I was particularly interested in the authors' studies of G3BP interaction alterations that accompanied overexpression of the N-protein peptide, this is a neat example of host protein network rewiring (Figure 4E). I'm curious if the authors attempted to overexpress the full-length N protein +/- mutation, and see how this alters G3BP1 interactions? (I'm not suggesting that this is necessary.)

By mapping the interaction between N protein and G3BP, the authors have certainly contributed to our understanding of SARS-CoV-2 host interactions. I'm curious if they have attempted to dock structures of N protein and G3BP, taking into account the linear binding motif they have identified. If this is possible, I'd be curious if any interesting insights may be revealed by such an exercise.

I suggest that the authors review the manuscript for grammatical errors and clarity. The figures are generally well composed, but some subfigures could be labeled better, particularly the X-axis on the volcano plots.

My specific suggested revisions are below.

-Several times in the manuscript, the authors refer to their approach as 'scalable', however

considering that their phage display interaction methodology requires the purification of dozens of recombinant proteins -- a procedure which can be time consuming -- the authors might reconsider their choice of words, or clarify how they intend to scale this approach for other applications they envision.

Page 4:

-The manuscript cites manuscripts describing ProP-PD technology, but a brief description of the screening pipeline would be helpful for the reader. It's also unclear to me what the authors mean by 'amino acid resolution of the binding sites,' is the RiboVD library tiled in single amino acid increments?

-The RiboVD library is generated against 229 RNA viruses, but I only saw coronavirus data in the manuscript and supplement (please correct me if I'm wrong). It would be helpful to clarify in the manuscript that only coronavirus data is being reported, and that the high-level descriptions of the results (such as at the top of page 5) are based only on the coronavirus portion of the RiboVD library. It would also be useful to mention if there any plans to publish the interaction data corresponding to non-coronavirus proteins in the library.

-If there was a particular logic / rationale for choosing specific coronavirus host interactors for recombinant protein production, it would be good to describe this.

Figure 2D:

It looks like the bait bands in the IP are not all of equal intensity, which is noteworthy considering the subtle differences in G3BP1 band intensity. The authors might consider quantifying the IP western blot bands and normalizing for bait intensity.

Subfigures 2I, 2J, 4E:

It would be helpful if the volcano plots in the manuscript were more clearly labeled on the X-axis so the reader can better understand which direction indicates an increase in signal for which condition.

It would be helpful if the authors included line numbers in future submissions, to permit reviewers to cite specific locations in the manuscript.

Thank you for the opportunity to review this manuscript.

Sincerely,
David E. Gordon

Reply to reviewers

We thank the reviewers for the helpful suggestions that have improved the quality of our manuscript. A point by point reply is provided below, with our responses in bold.

Reviewer #1 (Remarks to the Author):

Kruse et al describes the use of a large phage library displaying 16-mer peptides covering accessible regions of RNA virus proteins to identify short linear motif (SLiM) in these that can interact with a collection of 57 different protein interaction domains from 53 human proteins. This study reports 269 putative SLiM-based interactions between 44 human protein domains and 64 viral proteins from 18 coronavirus strains, focusing on those involving SARS-CoV-2, SARS-CoV, and MERS-CoV. Functional follow-up studies are described that address strong binding of an ØxFG motif in the nucleocapsid (N) proteins from SARS-CoV-2 and SARS-CoV to the NTF2 domain of the stress-granule protein G3BP1 and G3BP2. Additional ØxFG-containing host cell proteins binding to G3BP1/2 NTF2 domains are identified from a human peptide library. Further studies on viral and cellular ØxFG motifs suggest a mechanistic model for stress-granule disruption by SARS-CoV-2 and SARS-CoV N proteins, and show that blocking of this process has potential for antiviral development by potently interfering with SARS-CoV-2 replication.

These studies represent a very large and interesting body of work involving diverse experimental approaches. All work has been carried out in a highly professional manner, and the conclusions of the study are well supported by the results presented. A wealth of new information as well as a strong case for the utility of the proteomic peptide-phage display approach in further studies on virus-host interactions are provided. Thus, this study could be defined a landmark paper in its field even though none of the individual findings reported in it constitute a distinct breakthrough.

This study could obviously be extended into a number of directions, and it reports several observations that would be interesting to follow up in more detail. However, given the already extensive nature of this study, such work should probably be left for the future. On the other hand, the paper has no deficiencies or flaws that would clearly need to be addressed by additional experimentation before publication.

We thank the reviewer for the positive comments on our work

Major comments:

The experimental strategy and rationale of the study should be explained better:

1) A study of 1074 viral proteins from 229 RNA viruses is introduced to the reader, but all hits of the screen involve coronaviruses. Presumably also peptides from other viruses were selected by some of the 139 bait proteins used but were excluded from the current results. If so, this should be better explained.

Our response:

The reviewer correctly notes that many peptides from other RNA viruses were selected by the 139 bait proteins and that these results will be reported in future publications. We have clarified this in the revised manuscript.

And even if the rest of the hits will be reported in future publications, it would be of interest to know what percentage of all the discovered interactions are covered by the 269 coronavirus SLiM-based interactions now listed in Table S2.

Our response:

The 269 coronavirus SLiM-based interactions constitute 13% of all identified interactions. We have provided this information in the revised manuscript.

2) As pointed out by the authors the SLiM phage library approach can valuably complement more unbiased discovery approaches, such as large scale mass spectrometry or CRISPR-based screens. However, the 139 human bait proteins used cover only a tiny fraction of all relevant protein interaction domains (PID). This limitation should be discussed,

Our response:

We agree that maybe the major limitation/bottleneck of the phage library approach is the access to or production of PID baits. We have included these thoughts in the discussion in the revised manuscript.

and the current rationale of selecting the PIDs used as baits should be better explained It is mentioned that 57 were selected because they are present in proteins already reported as SARS-CoV-2 interactors, whereas 82 were selected based on some other logic that is not explained.

Our response:

Yes, the 57 PIDs were selected because they were previously reported to interact with SARS-CoV-2 encoded proteins. For this set of baits the aim was to identify novel SLiMs mediating these interactions.

There is no direct coronavirus based rationale for selecting the 82 PIDs. Rather these baits were chosen for accessibility to cDNA and efficient recombinant expression in our E. coli production pipeline. However, the 82 PIDs covers several well-known protein domain families of which many has previously been shown to bind viral proteins. With this set of baits, we aimed to uncover virus-host protein-protein interactions that have not been described before and at the same time identify the SLiMs mediating these interactions. As it turns out, we identified 165 out of the 269 interactions with this set of baits.

Furthermore, this suggests that any collection of PIDs could potentially be applied to our phage RiboVD library to identify novel interactions and SLiMs.

We have clarified this point in the revised manuscript.

It would also be of interest to know how many of the 269 SLiM-based hits were selected by these 57 vs. 82 baits.

Our response:

The numbers are 104 and 165 respectively. We have added these numbers to the revised manuscript.

3) Despite the space restrictions the essential bioinformatic basis and design of the large RiboVD library should be clarified instead of just referring to more extensive information in other publications and databases.

Our response:

We have provided a more extensive method section for the bioinformatic basis and design of the large RiboVD library in the revised manuscript.

4) It is understandable that due to the large collection of advanced methods used in this study the Materials & methods section has been written in a concise manner. Although the quality of the data looks excellent, the validity of many the experimental approaches, for example related to affinity measurement and virological assays, is therefore difficult to judge. A more detailed version Materials & methods-section, perhaps as a supplementary appendix would therefore be helpful.

Our response:

We have provided a more detailed version of the method section in the revised manuscript where needed as suggested by the reviewer.

Reviewer #2 (Remarks to the Author):

This study reported the use of the proteomic peptide-phage display (ProP-PD) platform to dissect the complex interactions of RNA viruses with host cellular proteins. This methodology is based on the documented interactions of short linear interaction motifs in viral proteins with the globular domains of cellular proteins. Thus, other forms of protein-protein interactions are not supposed to be covered by this method.

The authors have designed a ProP-PD library that covers 1074 viral proteins from 229 RNA viruses, including 19549 unique 16 amino acid long peptides. They went on and tested SARS-CoV-2 proteins against a total of 139 host protein domains that have been previously reported to interact with SARS-CoV-2 (57 domains) or other viruses (82 domains). 269 interactions were identified, 27 were validated by fluorescent polarization (FP) affinity assay.

The strength of this approach is to precisely determine the interacting motifs between the viral and cellular proteins. Since the pre-selected nature of the cellular protein domains, this method does not necessarily reveal novel virus-host interactions.

Our response:

We thank the reviewer for the positive comments. The approach can in principle also uncover novel virus-host interactions as well as identify the details of known interactions.

The authors went further to characterize the interactions of the FG motif in SARS-CoV-2 N protein and G3BP1, and showed the inhibitory effect of this short peptide, when fused with GFP, by more than 3-fold. A stronger binding motif FGDF from SFV NSP3 proteins exhibited a much greater inhibition by competitive binding to G3BP1, suggesting a potential therapeutic value of such inhibitory peptides.

More experiments were performed to understand how N protein modulates stress granule formation and how G3BP1 interacts with other cellular proteins, which led to the identification of new G3BP-interacting peptide motifs and new cellular partner proteins.

A few questions to be answered:

1. One aim of this study is to identify peptide-based antiviral drugs. Please discuss any precedent of using peptide as antivirals and the challenges compared to small molecule drugs.

Our response:

We have discussed this point in the revised discussion.

2. Please discuss and acknowledge the strength and limitation of ProP-PD approach in discovering new virus-host interactions.

Our response:

We have included a section on this in the discussion part of the revised manuscript as also suggested by the other reviewers.

3. Fig 2G: the signal of N protein in the IP is fairly weak, which questions the importance of this interaction in SARS-CoV-2 infection and the value as antiviral drugs.

Our response:

The interaction between SARS-CoV-2 N and G3BP1/2 is mediated by the Φ xFG short linear motif. We agree with the reviewer that this interaction is not a strong one compared to e.g. many domain-domain mediated interactions. This is an inherent property of most SLiM mediated protein-protein interactions and is the primary reason why this type of interaction can be technically challenging to detect using traditional methods for probing interactions such as immunoprecipitations (as used here). Nevertheless, many SLiM mediated protein-protein interactions are highly important and essential for the cell. We are convinced that the relatively weak N signal is due to technical limitations and that this is not predictive of the functional importance of the N-G3BP1 interaction.

4. Fig 3D, E: how did the authors distinguish early stage from late stage SARS-CoV-2 infected cells after 6 hours of infection? Would it be more rational to examine cells at different time points (for example, 4 hours vs 12 hours) to claim early vs late stage infections?

Our response:

We define early infection as very low N protein levels which is weakly stained, and late stage infection is defined as cells containing high levels of N protein. We did try analyzing cells at different time points after infection however, at 12 h there are always both highly infected cells from the first round of infection and very low level of infection grade due to second or third round of infection. At 4 hours it was very difficult to see any N protein staining and therefore not possible to distinguish uninfected and low infected cells. We therefore settled on 6 h because then we clearly see both low, high and non-infected cells as a representative view.

In Fig 3D, visually, all infected cells appear to have G3BP1 granules.

Our response:

We respectfully disagree with the reviewer on this point. Please look at the zoom in of panel 1-3. The cell shown in panel 3 has no stress granules (G3BP1 staining in green).

Reviewer #3 (Remarks to the Author):

Article summary:

The authors set out to identify viral short linear interaction motifs (SLiMs) that interact with human proteins, with a focus on coronaviruses. The authors began by generating a phage display library of short peptides identified from the predicted unstructured regions of RNA virus proteins (including but not limited to coronaviruses), and utilized this library for in-vitro binding experiments with 57 domains from known human SARS-CoV-2 interacting proteins as well as 82 other human peptide binding protein domains. Reassuringly, the resulting interaction network included known interactions between coronavirus proteins and host proteins, such as interactions between N protein and G3BP1/2, as well as between Nsp14 and AP2.

The authors focused on the linear motif phi-x-F-G in coronavirus N protein and its interaction with the stress granule associated proteins G3BP1/2. They evaluated peptide-based therapeutic interventions for SARS-CoV-2 infection, theorizing such perturbations would interfere with the binding of this linear motif and G3BP and inhibit infection, and successfully verified this hypothesis. They further performed biochemical and cell biological studies to gain insights into the functional and biochemical consequences of the interaction between N protein and G3BP1/2 through the N protein phi-x-F-G motif.

Applying mutations in the phi-x-F-G motif to N protein, they demonstrate convincingly that the wild type motif is required to interfere with stress granule assembly (Figure 3B). The authors also demonstrate that overexpression of the N protein peptide displaces a number of cellular G3BP interactions, raising interesting questions regarding host network rewiring resulting from the N protein – G3BP interaction.

My overall impressions and suggestions:

Systematic mapping of interactions between viral SLiMs and host proteins is a helpful contribution towards our structural understanding of host-pathogen interactions. As the authors rightly point out, their approach builds upon large-scale interactome studies by providing further detail regarding interaction interfaces. This information is useful for the design of small molecule inhibitors, and the authors prove this point by demonstrating a peptide-based inhibitor of coronavirus infection. The authors' methodology and conclusions are sound and are supported by their data.

We thank the reviewer for the positive comments on our work.

I was particularly interested in the authors' studies of G3BP interaction alterations that accompanied overexpression of the N-protein peptide, this is a neat example of host protein network rewiring (Figure 4E). I'm curious if the authors attempted to overexpress the full-length N protein +/- mutation, and see how this alters G3BP1 interactions? (I'm not suggesting that this is necessary.)

Our response:

We did indeed try as a first attempt to overexpress the full-length N protein +/- mutation to see how this altered the G3BP1 interaction pattern. However, we had difficulties obtaining uniform and sufficiently high expression of N proteins in the cell cultures so the subsequent MS analysis turned out to be inconclusive. We therefore turned to the N-peptide approach that gave very reliable and robust results.

By mapping the interaction between N protein and G3BP, the authors have certainly contributed to our understanding of SARS-CoV-2 host interactions. I'm curious if they have attempted to dock structures of N protein and G3BP, taking into account the linear binding motif they have identified. If this is possible, I'd be curious if any interesting insights may be revealed by such an exercise.

Our response:

We were also interested in getting a more detailed molecular understanding of the N-G3BP interaction interface. To this end, we tried to solve the structure of G3BP1 bound to a peptide containing the phi-x-F-G motif using x-ray crystallography. Unfortunately, this attempt was unsuccessful.

We never tried to dock the N protein onto G3BP structures.

I suggest that the authors review the manuscript for grammatical errors and clarity. The figures are generally well composed, but some subfigures could be labeled better, particularly the X-axis on the volcano plots.

Our response:

We have gone through the manuscript and figures as suggested by the reviewer.

My specific suggested revisions are below.

-Several times in the manuscript, the authors refer to their approach as 'scalable', however considering that their phage display interaction methodology requires the purification of dozens of recombinant proteins -- a procedure which can be time consuming -- the authors might reconsider their choice of words, or clarify how they intend to scale this approach for other applications they envision.

Our response:

We agree with the reviewer and have removed the word "scalable".

We agree that production of recombinant protein baits for the phage display screens may constitute a considerable work-load and is one of the major limitations of this technology (especially if you want to screen many baits). We have pointed this out clearly in the revised discussion.

Page 4:

-The manuscript cites manuscripts describing ProP-PD technology, but a brief description of the screening pipeline would be helpful for the reader. It's also unclear to me what the authors mean by 'amino acid resolution of the binding sites,' is the RiboVD library tiled in single amino acid increments?

Our response:

We have included a brief description of the screening pipeline in the revised manuscript as suggested by the reviewer.

We agree with the reviewer and have changed "amino acid resolution" to "high resolution".

-The RiboVD library is generated against 229 RNA viruses, but I only saw coronavirus data in the manuscript and supplement (please correct me if I'm wrong). It would be helpful to clarify in the manuscript that only coronavirus data is being reported, and that the high-level descriptions of the results (such as at the top of page 5) are based only on the coronavirus portion of the RiboVD library. It would also be useful to mention if there any plans to publish the interaction data corresponding to non-coronavirus proteins in the library.

Our response:

This is correct. In the current manuscript we focus on the peptides identified from the coronavirus part of the RiboVD library. In addition, many peptides from the other RNA viruses were selected by the 139 bait proteins. These results will be reported in future publications. We have clarified this in the revised manuscript.

-If there was a particular logic / rationale for choosing specific coronavirus host interactors for recombinant protein production, it would be good to describe this.

Our response:

57 protein domains (PIDs) were selected because they were previously reported to interact with SARS-CoV-2 encoded proteins. Furthermore, these baits expressed well in our E. coli production system. For this set of baits the aim was to identify novel SLiMs mediating these interactions.

There is no direct coronavirus based rationale for selecting the remaining 82 PIDs. Rather these baits were chosen for accessibility to cDNA and efficient recombinant expression in our E. coli production pipeline. However, the 82 PIDs covers several well-known protein domain families of which many have previously been shown to bind viral proteins. With this set of baits, we aimed to uncover viral-human protein-protein interactions that have not been described before and at the same time identify the SLiMs mediating these interactions. As it turns out, we identified xx interactions and xx putative SLiMs with this set of baits.

Furthermore, this suggests that any collection of PIDs could potentially be applied to our phage RiboVD library to identify novel viral-host interactions and SLiMs.

We have clarified this point in the revised manuscript.

Figure 2D:

It looks like the bait bands in the IP are not all of equal intensity, which is noteworthy considering the subtle differences in G3BP1 band intensity. The authors might consider quantifying the IP western blot bands and normalizing for bait intensity.

Our response:

Using LI-COR software we have quantified the G3BP1 band and normalized to bait intensity as suggested by the reviewer in an up-dated figure 2D. This reveals a 4-5 fold difference between the G3BPi wt and ctrl situation.

Subfigures 2I, 2J, 4E:

It would be helpful if the volcano plots in the manuscript were more clearly labeled on the X-axis so the reader can better understand which direction indicates an increase in signal for which condition.

Our response:

We have labeled more clearly the volcano plots on top to make them more accessible to the reader. We prefer to keep x-axis label as is.

It would be helpful if the authors included line numbers in future submissions, to permit reviewers to cite specific locations in the manuscript.

Our response:

Done.

Thank you for the opportunity to review this manuscript.

Sincerely,

David E. Gordon

Reviewers' Comments:

Reviewer #2:

Remarks to the Author:

Comments for Figure 3d: It is clear that the mock cells do not show any stress granules. But cells shown in panels 1 to 3, stained positive for CoV-2 N protein, all have stress granules (G3BP1 punctates), just vary in numbers. Although panel 3 shows only one stress granule, it would be mistaken if the cell in panel 3 was scored as stress granule negative.

I trust it is a better way to calculate the potential correlation between the number of stress granules vs the level of N protein, rather than the % of cells with stress granules vs N protein level as shown in figure 3e.

Low level of N does not necessary indicate early stage of infection, as the cell population is heterogeneous, some cells are less permissive to SARS-CoV-2 infection than others, and may express less N and produce fewer virus particles. It is more straight forward to assess the stress granule formation and N protein expression level, and this is not expected to change the main conclusion of the study.

Reviewer #3:

Remarks to the Author:

If only coronavirus data is covered in this manuscript, it may be best not to mention the larger scope of the library in the abstract of the paper, as this could confuse readers. Likewise, in the introduction it should be clear and up-front that only coronavirus proteins are studied in the current manuscript.

The example of the T20 inhibitor referenced in the Discussion is not ideal, since T20 is a fusion inhibitor, and does not need to cross membranes to elicit an effect. I suggest the authors identify an example of a peptide-based drug which is cell permeable, as this would better support their work and conclusions.

Apart from the above suggestions, my earlier concerns have been addressed. Thank you for the opportunity to review this manuscript.

Sincerely,
David Gordon

REVIEWERS' COMMENTS

Reviewer #2 (Remarks to the Author):

Comments for Figure 3d: It is clear that the mock cells do not show any stress granules. But cells shown in panels 1 to 3, stained positive for CoV-2 N protein, all have stress granules (G3BP1 punctates), just vary in numbers. Although panel 3 shows only one stress granule, it would be mistaken if the cell in panel 3 was scored as stress granule negative.

I trust it is a better way to calculate the potential correlation between the number of stress granules vs the level of N protein, rather than the % of cells with stress granules vs N protein level as shown in figure 3e.

Low level of N does not necessary indicate early stage of infection, as the cell population is heterogeneous, some cells are less permissive to SARS-CoV-2 infection than others, and may express less N and produce fewer virus particles. It is more straight forward to assess the stress granule formation and N protein expression level, and this is not expected to change the main conclusion of the study.

Our response:

We have now calculated the correlation between N protein levels and stress granules showing that in cells with low levels we detect numerous stress granules while at high levels we see few (new plot in fig. 3E and Supplementary Fig. 3f). We have removed any claims that low levels of N correspond to early infection and high levels of N corresponds to late infection. The text for this section has been modified to:

We next analyzed G3BP1 foci formation and cellular localization of viral dsRNA in relation to N protein expression levels in VeroE6 cells after six hours of SARS-CoV-2 infection (Fig. 3d,e). At this timepoint, a mixture of early and later stage infected cells is observed. In mock treated cells we detected no cells with more than two G3BP1 foci and based on this we set the background threshold at three G3BP1 foci per cell (Supplementary Fig. 3f). In infected cells with low levels of N protein (below 10000 fluorescent units) a large proportion of cells had multiple G3BP1 foci (Fig. 3e). In cells with low levels of N, this protein and viral dsRNA co-localized with G3BP1 to stress granules (Fig. 3d). However, in cells with high levels of N protein only 2 out of 11 cells had G3BP1 foci above threshold levels. Collectively our results suggest that low levels of N protein are insufficient to disrupt stress granule formation and instead N and viral dsRNA co-localize with G3BP1 in these structures.

Reviewer #3 (Remarks to the Author):

If only coronavirus data is covered in this manuscript, it may be best not to mention the larger scope of the library in the abstract of the paper, as this could confuse readers. Likewise, in the introduction it should be clear and up-front that only coronavirus proteins are studied in the current manuscript.

Our response:

We have adjusted the manuscript to reflect this.

The example of the T20 inhibitor referenced in the Discussion is not ideal, since T20 is a fusion inhibitor, and does not need to cross membranes to elicit an effect. I suggest the authors identify an example of a peptide-based drug which is cell permeable, as this would better support their work and conclusions.

Our response:

We have adjusted the discussion to point out that T20 is a fusion inhibitor and have provided information on the current status of cell permeable peptides in the clinic (wording as agreed in email correspondence with editor).

Apart from the above suggestions, my earlier concerns have been addressed. Thank you for the opportunity to review this manuscript.